# Material category of visual objects computed from specular image structure

Alexandra C. Schmid ●[1] ✉, Pascal Barla ●[2] & Katja Doerschner ●[1]

Recognizing materials and their properties visually is vital for successful interactions with our environment, from avoiding slippery floors to handling fragile objects. Yet there is no simple mapping of retinal image intensities to physical properties. Here, we investigated what image information drives material perception by collecting human psychophysical judgements about complex glossy objects. Variations in specular image structure—produced either by manipulating reflectance properties or visual features directly—caused categorical shifts in material appearance, suggesting that specular reflections provide diagnostic information about a wide range of material classes. Perceived material category appeared to mediate cues for surface gloss, providing evidence against a purely feedforward view of neural processing. Our results suggest that the image structure that triggers our perception of surface gloss plays a direct role in visual categorization, and that the perception and neural processing of stimulus properties should be studied in the context of recognition, not in isolation.

Our visual experience of the world arises from light that has been reflected, transmitted or scattered by surfaces. From this light we can tell whether a surface is light or dark, shiny or dull, translucent or opaque, made from platinum, plastic or pearl (Fig. 1a). The quick and accurate recognition of materials and their intrinsic properties is central to our daily interactions with objects and surfaces, from inferring tactile information (is it smooth, heavy, soft?) to assessing function (is it edible, fragile, valuable?). Yet material perception is not trivial because the structure, spectral content and amount of light reaching our eyes depend not only on surface reflectance, transmittance and scattering properties, but also on complex interactions with the three-dimensional (3D) shape, position and orientation of surfaces with respect to each other, light sources and the observer. Thus, our ability to effortlessly discern a wide variety of materials that each have a potentially unlimited optical appearance is remarkable, and serves as a compelling example of an important but unresolved challenge in visual neuroscience: how does the brain disentangle the conflated factors contributing to the retinal image to perceive our visual world?

Although there is a growing body of work investigating the visual perception of material properties such as colour, lightness, transparency, translucency and gloss[1–6], there is comparatively little investigating the recognition of different material classes such as plastic, pearl, satin, steel and so on[7–17]. For example, previous research has discovered a limited set of image conditions (photogeometric constraints) that trigger the perception of a glossy versus a matte surface, involving the intensity, shape, position and the orientation of specular highlights (bright reflections[18–24]) and lowlights (dark reflections[25]) with respect to diffuse shading. However, it remains unknown what image information triggers our perception of different materials. A possible reason for the disparate focus on material properties versus classes is that studying properties like colour and gloss seems more tractable than discovering the necessary and sufficient conditions for recognizing the many material classes in our environment. The challenge is that the perceptual space of materials is unspecified; there are many different optical 'appearances' that can look like steel (think polished, scratched, rusted), or plastic (smooth and glossy or rough and dull).

Furthermore, a traditional feedforward view of neural processing is often assumed in which the recognition of objects and materials proceeds from the processing of low-level sensory information (image features) to the estimation of shape and surface properties (often referred to as mid-level vision[2,3]) to the high-level recognition of object and material categories[26] (feedforward hypothesis; Fig. 1b, top).

[1]Department of Psychology, Justus Liebig University Giessen, Giessen, Germany. [2]Inria, Bordeaux, France. ✉e-mail: Alexandra.Schmid@nih.gov

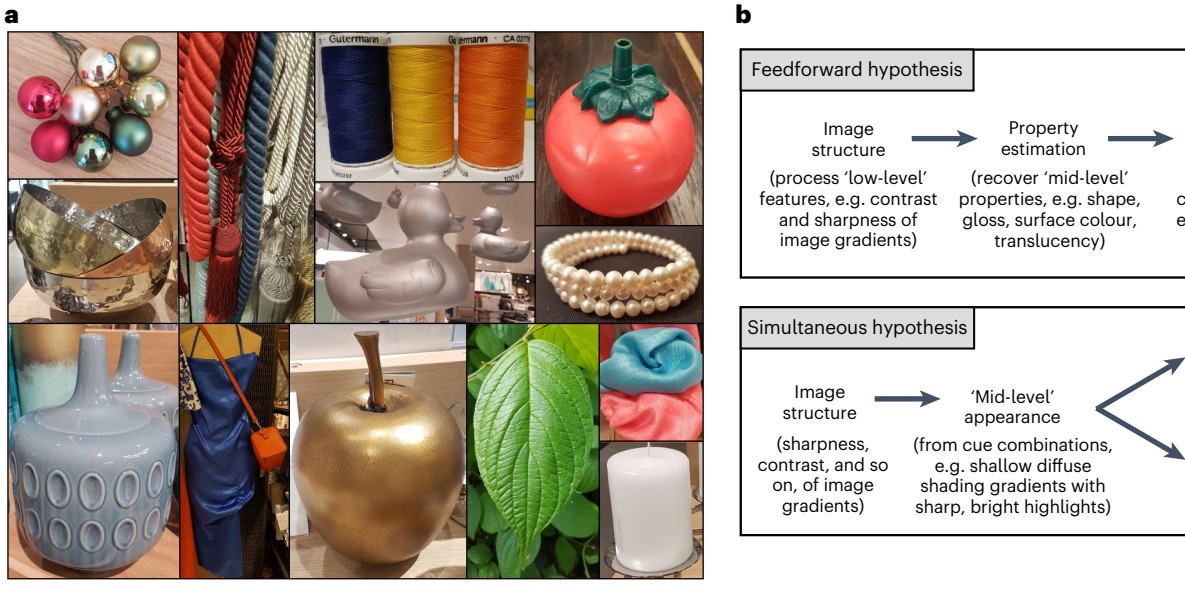

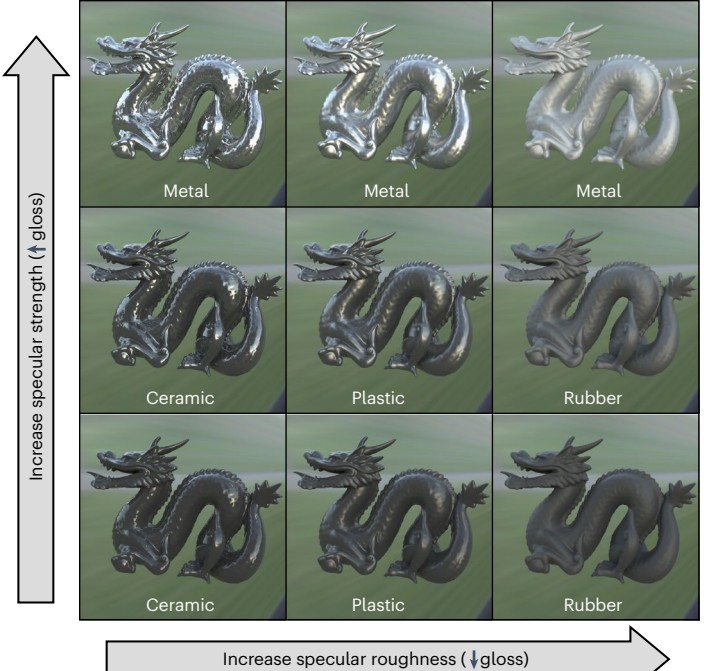

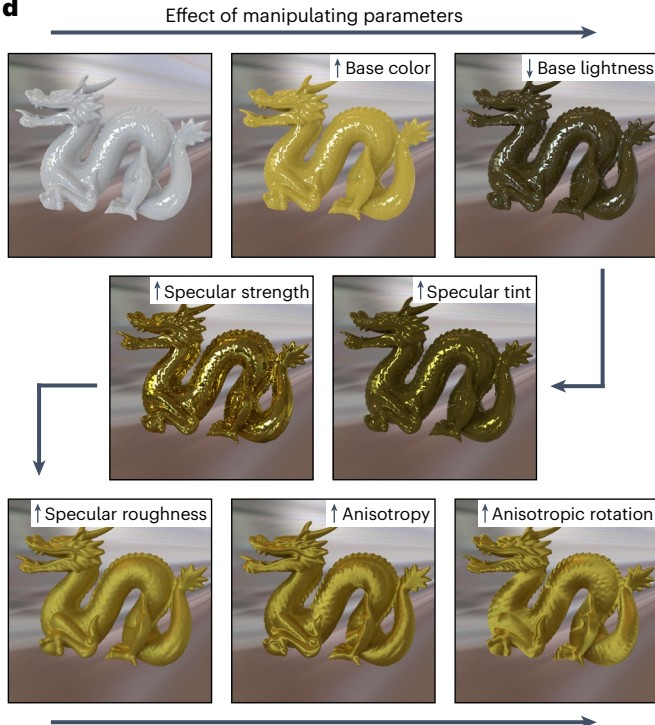

**Fig. 1 | The relationship between reflectance properties, image structure and material appearance. a**, Most objects that we see every day are made from materials whose specular reflectance properties produce a characteristic image structure. The appearance of these reflections is determined (and constrained) by the way in which these materials scatter light, in addition to other generative processes (Fig. 2). **b**, However, the extent and mechanism by which specular structure contributes to categorization remains unknown. A feedforward view of neural processing (top) assumes that categories are defined by combinations of estimated mid-level properties like gloss, colour and apparent shape, which the visual system tries to 'recover' from the image[67]. By contrast, the simultaneous hypothesis (bottom) assumes that the visual system learns naturally about statistical variations (regularities) in image structure, from which the identity or category of a material can be 'read out' simultaneously with surface qualities like gloss[29]. **c**, Some aspects of a surface's appearance (such as perceived gloss) tend to correlate with physical properties (such as specular reflectance), all else being equal. Systematically

manipulating the specular reflectance properties of otherwise identical objects causes salient visual differences in the appearance of highlights (Fig. 2), which affects perceived gloss in predictable ways. Specifically, increasing specular strength (bottom to top) increases the contrast of specular highlights, causing the dragon to appear more glossy; increasing specular roughness (from left to right) decreases the clarity (or sharpness) of the specular highlights, causing the dragon to appear less glossy. These manipulations also affect our qualitative (categorical) impressions: the surfaces resemble different materials like glazed ceramic, glossy plastic, dull plastic, rubber, polished metal and brushed metal. Because shape, surface colour and illumination conditions are held constant, all visual differences are caused by variations in specular reflectance properties, suggesting that specular structure may directly contain diagnostic information about material class, simultaneously with surface gloss (note that for some materials like gold, colour information also contributes to perceptual classification). **d**, Reflectance parameters manipulated in the experiments, and examples of how this affected the visual appearance of surfaces.

Within this framework it would make sense to first study how the visual system computes mid-level material properties like surface gloss and colour from images, because material class is thought to be subsequently computed from the high-dimensional feature space defined by these component mid-level properties[9–12,27,28].

An alternative view suggests that the visual system does not try to recover physical surface properties like diffuse reflectance (seen as surface colour) and specular reflectance (seen as gloss) per se, but rather learns about statistical variations or regularities in image structure from which both material properties like gloss and material categories like plastic can be 'read out' simultaneously[5,6,29,30] (simultaneous hypothesis; Fig. 1b, bottom). For example, in Fig. 1c, systematic differences in the surface reflectance properties of otherwise identical objects produce variations in the appearance of specular highlights, such as their contrast, sharpness and coverage (Fig. 2). These variations in highlight appearance affect not only the quantitative level of surface gloss perceived (shiny to dull), but also the quality—or material category—of the surface altogether (rubber, plastic, ceramic or metal), and it is possible that the same visual features, or cues, directly underlie both processes. At present, however, the relationship between image structure, perceptual properties like surface gloss and material recognition is unclear.

Here, we test the precise role of image structure produced by specular reflections in material recognition by rendering complex but carefully controlled stimuli in natural illumination fields, measuring and manipulating aspects of specular image structure, and collecting human psychophysical judgements about surface appearance in a series of experiments. Our results show that the visual system is highly sensitive to specular structure for material recognition, even in the absence of information from other physical sources like surface texture, transmittance and subsurface scattering, suggesting that specular reflections play a more extensive role in visual recognition than previously appreciated. Furthermore, the data reveal that, rather than materials being derived via the estimation of material properties like gloss (feedforward hypothesis), it is more probable that visual cues for gloss perception are constrained by material class, implying that the perception and neural processing of gloss and other stimulus properties should be studied in the context of recognition rather than in isolation. Moreover, our results demonstrate that material category is directly computable from measurable visual features of specular structure, and that manipulating these features transforms perceived category, suggesting that specular structure provides direct diagnostic information about an object's material. We discuss a simultaneous account of material perception (simultaneous hypothesis) in which stimulus properties like gloss are co-computed with (constrained by) our qualitative holistic impressions.

## Results

### Specular appearance yields a wide range of material classes

If the visual system is sensitive to specular reflections for material recognition beyond whether a surface is shiny or matte[24], then altering the appearance of specular reflections should lead to changes in perceived material. To test this, we computer-rendered glossy objects with different surface reflectance properties under natural illumination (Fig. 1d). We parametrically manipulated base colour (lightness and saturation of the diffuse component) in addition to five specular reflection parameters (specular strength, specular tint, specular roughness, anisotropy and anisotropic rotation) to control the appearance of specular reflections with respect to diffuse shading, resulting in 270 stimuli (Supplementary Fig. 1). We collected unbiased participant-generated category terms for each stimulus using a free-naming task (Experiment 1) in which participants ($n = 15$) judged what material each object was made from with no restrictions.

After processing for duplicates and similar terminology (Analyses), more than 200 terms were used to describe the materials, over half of which were category terms (nouns such as porcelain, gold, plastic, stone, ceramic, chocolate, pearl, soap, wax, metal, bronze, rubber, fabric, velvet; Supplementary Figs. 2 and 3). Although the use of each term was well distributed among participants (that is, category labels did not come from the same few participants), semantic labels are only relevant to the extent that they capture qualitative perceptual differences in visual material appearance. For example, dark brown stimuli with medium-clarity specular reflections might be labelled as 'melted chocolate' or 'mud' but would be qualitatively visually equivalent to one another. Such stimuli would have a different visual quality to those with low-clarity, dim reflections, which might be labelled as 'latex' or 'rubber' (Fig. 3a). Therefore, we sought to reveal the latent perceptual space of materials for our stimulus set, with the following steps.

First, we reduced the set of category labels generated from the free-naming task to those that were used by at least five participants, and merged visually or semantically similar terms, guided by correlations between the categories (Analyses and Supplementary Fig. 4). The reduced set of 18 category terms is shown in Fig. 3b. We decided to not reduce the number of category terms further to offer a wide range of choices to subjects in the next experiment.

Second, a separate set of participants ($n = 80$) completed a multiple-alternative forced-choice task (18-AFC task; Experiment 2) in which they were asked to choose the material category that best applied to each stimulus (Supplementary Fig. 5). For this experiment, the stimulus set was extended to include a larger range of reflectance parameters, two shapes (dragon and bunny) and two lighting environments (kitchen and campus), resulting in 924 stimuli (Supplementary Fig. 1). Participants ($n = 20$ per stimulus) provided confidence ratings (converted to a score between 1 and 3), which allowed them to indicate their satisfaction with the category options presented. The confidence ratings for each category for each stimulus were summed across all participants, providing a distribution of category responses for each stimulus (category profiles) and a distribution of stimuli for each category (stimulus profiles) (Fig. 3b). For many stimuli, more than one category term applied; for example, Stimulus 1 was almost equally classified as 'glazed ceramic' and 'covered in wet paint', whereas Stimulus 6 was classified as both latex/rubber and plastic (Fig. 3b). This might be due to redundancies in the terminology and/or imperfect category membership driving different decision boundaries (for example, 'it looks a bit like plastic but also a bit like rubber'). Indeed, we found that the stimulus profiles for some of the categories correlated with one another (Supplementary Fig. 6).

Third, the data were subjected to a factor analysis to extract the common variance between categories and reveal orthogonal perceptual dimensions for our stimulus set. Figure 4a shows that there was no clear plateau in shared variance explained with each additional factor, and 12 factors were needed to account for at least 80% of the shared variance between stimuli (the upper limit is based on degrees of freedom; Analyses). Figure 4b shows example stimuli from the emergent dimensions, which were highly interpretable (12 plus one dimension that emerged from the negative loadings; Fig. 4c). Retaining eight or ten factors accounted for only approximately 60% and 70% of the common variance between categories, respectively, demonstrating that changing the appearance of specular reflections yields a diverse range of perceived materials that cannot be reduced to a small number of perceptual dimensions.

Surprisingly, the materials that were perceived extended beyond those expected based on the reflectance function used to generate them. In the real world, materials like porcelain, pearl, soap, wax, velvet and many others produce an image structure that is caused by the extent and way in which they transmit, internally scatter and disperse light in addition to pigment variations and mesoscale details like fibres in fabrics or scratches in anisotropic metals. Yet, participants reported seeing these materials for surfaces that were only defined by (uniform) diffuse and specular reflectance properties, despite the absence of

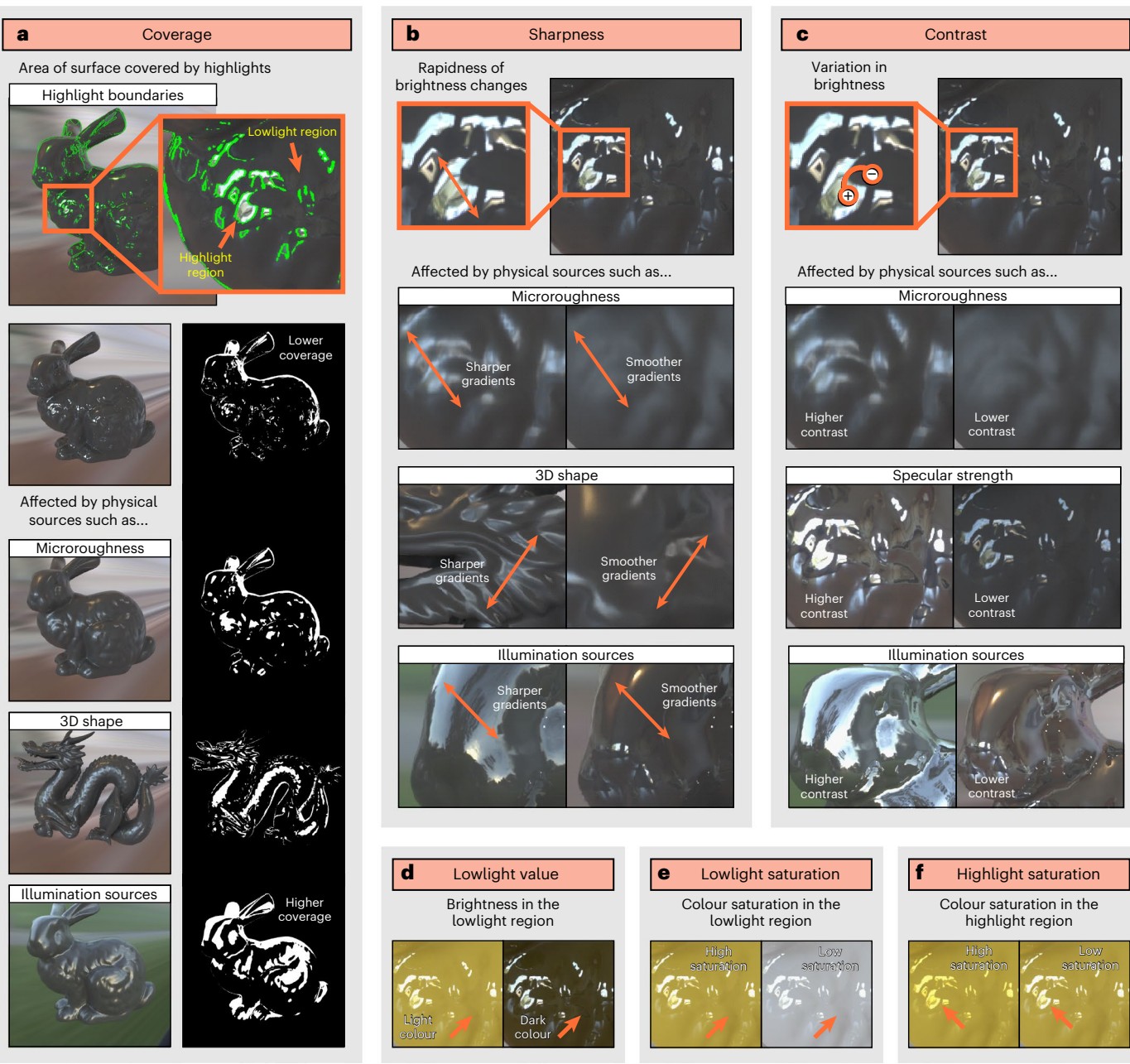

**Fig. 2 | Visual features linked with specular image structure. a–c**, The surface appearance of glossy objects is constrained by generative processes, which determine the coverage (**a**), sharpness (**b**) and contrast (**c**) of specular highlights. Specifically, image structure is constrained by the way that surface reflectance properties interact with 3D surface geometry and the light field[31]. Microroughness: whereas matte surfaces scatter light in all directions, glossy surfaces reflect light directionally, preserving the structure of the illumination field. Highly smooth surfaces produce narrow specular lobes[68] that increase the contrast and clarity (sharpness) of the reflected image relative to microscopically rougher surfaces, which produce broader specular lobes that blur this structure. Surfaces with higher microroughness also cause more spread-out specular highlights (higher coverage) owing to the larger range of surface normal orientations reflecting bright light sources towards the observer. Specular strength: whereas dielectric materials like plastic reflect only a proportion of light specularly, metals reflect all light specularly, increasing the contrast of the reflected image. Illumination sources: because of the structure-preserving properties of glossy objects, the contrast, sharpness and coverage of specular highlights depend on the incident light that is being reflected; that is, the intensity and structure of the illumination field. Three-dimensional shape: furthermore, 3D shape distorts this illumination structure. Specular reflections cling to points of high surface curvature and are elongated along (but slide rapidly across) directions of minimal surface curvature[23,24,69,70]. This means that 3D shape and observer viewpoint affect the location and distortion of specular reflections and thus the proximal stimulus properties of coverage, sharpness and contrast of the highlights. **d–f**, The brightness (**d**) and colour saturation inside (**e**) and outside (**f**) the highlight region are also determined by interactions between a surface's absorption/reflectance properties, 3D shape and the illumination field. For example, specular reflections from dielectric materials like plastic preserve the spectral content of incident light; hence they look tinted only if the prevailing illumination is itself coloured. By contrast, coloured metals affect the tint of reflected light. We refer to **a–c** as gloss cues and **d–f** as colour cues.

these other potentially diagnostic sources of image structure. Thus, our results reveal that the human visual system is highly sensitive to the image structure produced by specular reflections for material recognition, even for complex materials. Note that some categories (like gold metals) occurred because of our arbitrary choice of yellow for the coloured stimuli, and the outcome would be different for other

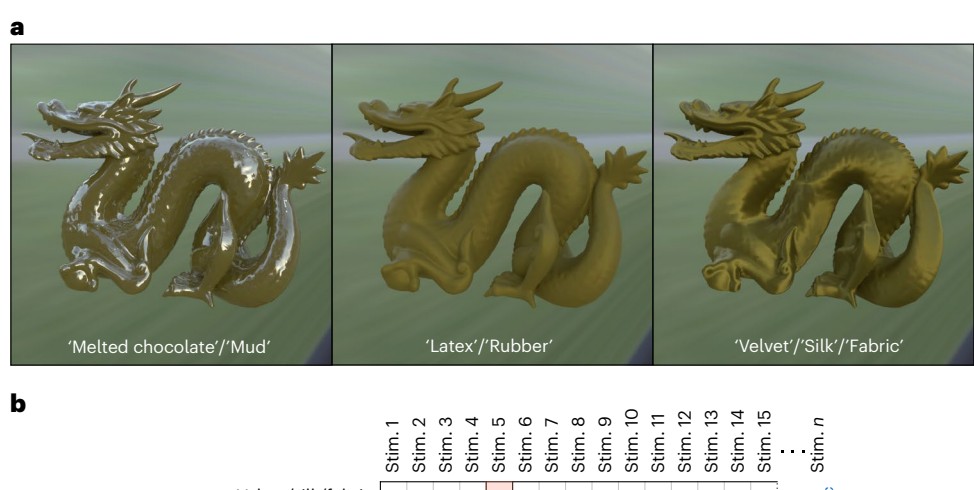

'Melted chocolate'/'Mud'   'Latex'/'Rubber'   'Velvet'/'Silk'/'Fabric'

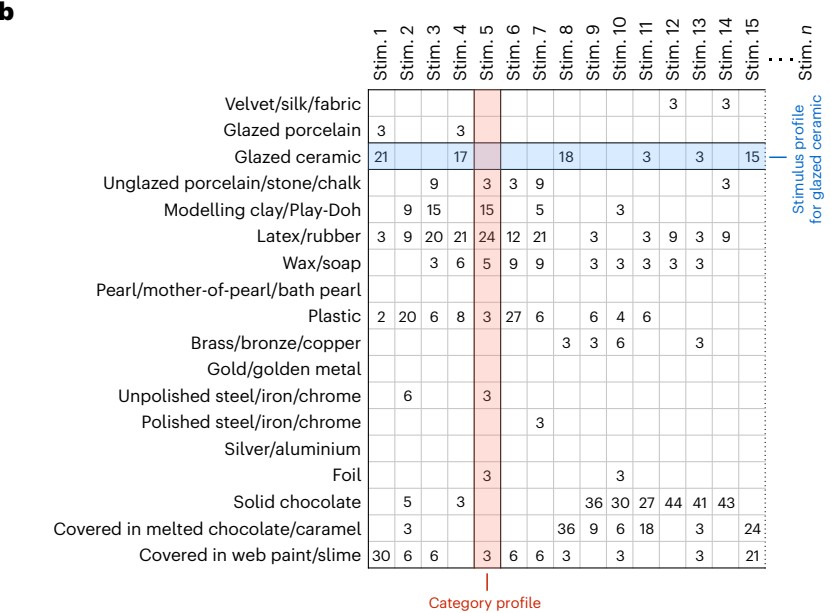

| | Stim. 1 | Stim. 2 | Stim. 3 | Stim. 4 | Stim. 5 | Stim. 6 | Stim. 7 | Stim. 8 | Stim. 9 | Stim. 10 | Stim. 11 | Stim. 12 | Stim. 13 | Stim. 14 | Stim. 15 | ... Stim. n |
|---|---|---|---|---|---|---|---|---|---|---|---|---|---|---|---|---|
| Velvet/silk/fabric | | | | | | | | | | | | 3 | | 3 | | |
| Glazed porcelain | 3 | | | 3 | | | | | | | | | | | | |
| Glazed ceramic | 21 | | | 17 | | | | 18 | | | 3 | | 3 | | | 15 |
| Unglazed porcelain/stone/chalk | | | 9 | | 3 | 3 | 9 | | | | | | | 3 | | |
| Modelling clay/Play-Doh | | 9 | 15 | | 15 | | 5 | | | 3 | | | | | | |
| Latex/rubber | 3 | 9 | 20 | 21 | 24 | 12 | 21 | | 3 | | 3 | 9 | 3 | 9 | | |
| Wax/soap | | | 3 | 6 | 5 | 9 | 9 | | 3 | 3 | 3 | 3 | 3 | | | |
| Pearl/mother-of-pearl/bath pearl | | | | | | | | | | | | | | | | |
| Plastic | 2 | 20 | 6 | 8 | 3 | 27 | 6 | | 6 | 4 | 6 | | | | | |
| Brass/bronze/copper | | | | | | | | 3 | 3 | 6 | | | 3 | | | |
| Gold/golden metal | | | | | | | | | | | | | | | | |
| Unpolished steel/iron/chrome | | 6 | | | 3 | | | | | | | | | | | |
| Polished steel/iron/chrome | | | | | | | 3 | | | | | | | | | |
| Silver/aluminium | | | | | | | | | | | | | | | | |
| Foil | | | | | 3 | | | | | 3 | | | | | | |
| Solid chocolate | | 5 | | 3 | | | | | 36 | 30 | 27 | 44 | 41 | 43 | | |
| Covered in melted chocolate/caramel | | 3 | | | | | | | 36 | 9 | 6 | 18 | | 3 | | 24 |
| Covered in web paint/slime | 30 | 6 | 6 | | 3 | 6 | 6 | 3 | | 3 | | | 3 | | | 21 |

Stimulus profile for glazed ceramic

Category profile for stimulus 5

**Fig. 3 | Multiple semantic terms can describe the same qualitative visual appearance of a material. a**, The object in the first panel has a dark brown, lumpy surface with medium-clarity specular reflections and could be labelled 'melted chocolate' or 'mud'. This surface has a different visual quality from the object in the middle panel, which has low-clarity, dim reflections that make it look like 'latex' or 'rubber'. The object in the third panel has very rough, anisotropic specular reflections that rapidly change in brightness at the boundaries between highlights and lowlights, giving it the visual characteristics of 'velvet', 'silk' or 'fabric'. **b**, Sum of confidence ratings from the 18-AFC experiment (Experiment 2) for each stimulus (Stim.) and each category for the first 15 stimuli (of 924).

Category profiles are the distribution of category responses for each stimulus and reveal the extent to which multiple category terms apply to the same stimulus. Stimulus profiles are the distribution of stimuli that were allocated to each category and reveal the extent to which category terms are correlated with one another. Stimulus profiles were used to reveal the latent perceptual space of materials for our stimuli (factor analysis; Fig. 4), and category profiles were used to calculate material dissimilarity scores between each pair of stimuli (representational similarity analysis; Supplementary Analysis B and Supplementary Fig. 15).

colours. Nevertheless, as we show later, such materials are additionally defined by specular image structure, and colour information alone is not sufficient to discriminate between materials.

**Material class is not determined by but may constrain gloss**
Although manipulating surface reflectance properties changed image structure in a way that participants interpreted as different materials, how this image structure relates to material recognition and the underlying mechanism is unclear. A feedforward approach assumes that the appearance of specular reflections determines surface glossiness, which in turn combines with other estimated mid-level properties (such as object shape, colour, translucency) to define the material category (Fig. 1b, top). If this is true, then we should be able to identify visual features (cues) that predict gloss perception, and the material categories from Experiment 2 should be associated with a particular level of gloss. We tested this in Experiment 3 in which a separate group of participants ($n = 22$) rated the perceived glossiness (gloss level)

of each of the 924 stimuli from Experiment 2. We directly measured visual features of the stimuli that describe the appearance of specular reflections and are based on generative constraints on how light is reflected and scattered by a surface's reflectance properties (Fig. 2). Three of these features—coverage, sharpness and contrast—have previously been found to predict participants' judgements of surface glossiness[31,32], so we refer to them as gloss cues. However, whereas Marlow and colleagues[31,32] used perceptual judgements of each cue to predict gloss, here we operationalised the cues using objective, image-based measures computed from dedicated render outputs (Analyses and Supplementary Fig. 9). Intuitively, coverage is the extent to which an object's surface is covered in specular highlights (bright reflections; Fig. 2a); sharpness refers to the rapidness of change between brighter and dimmer regions within and at the boundary of those highlights, and is usually related to the distinctness of the reflections (that is, the clarity of the reflected environment; Fig. 2b); and contrast is the variance in bright and dim regions caused by specular reflections, and usually

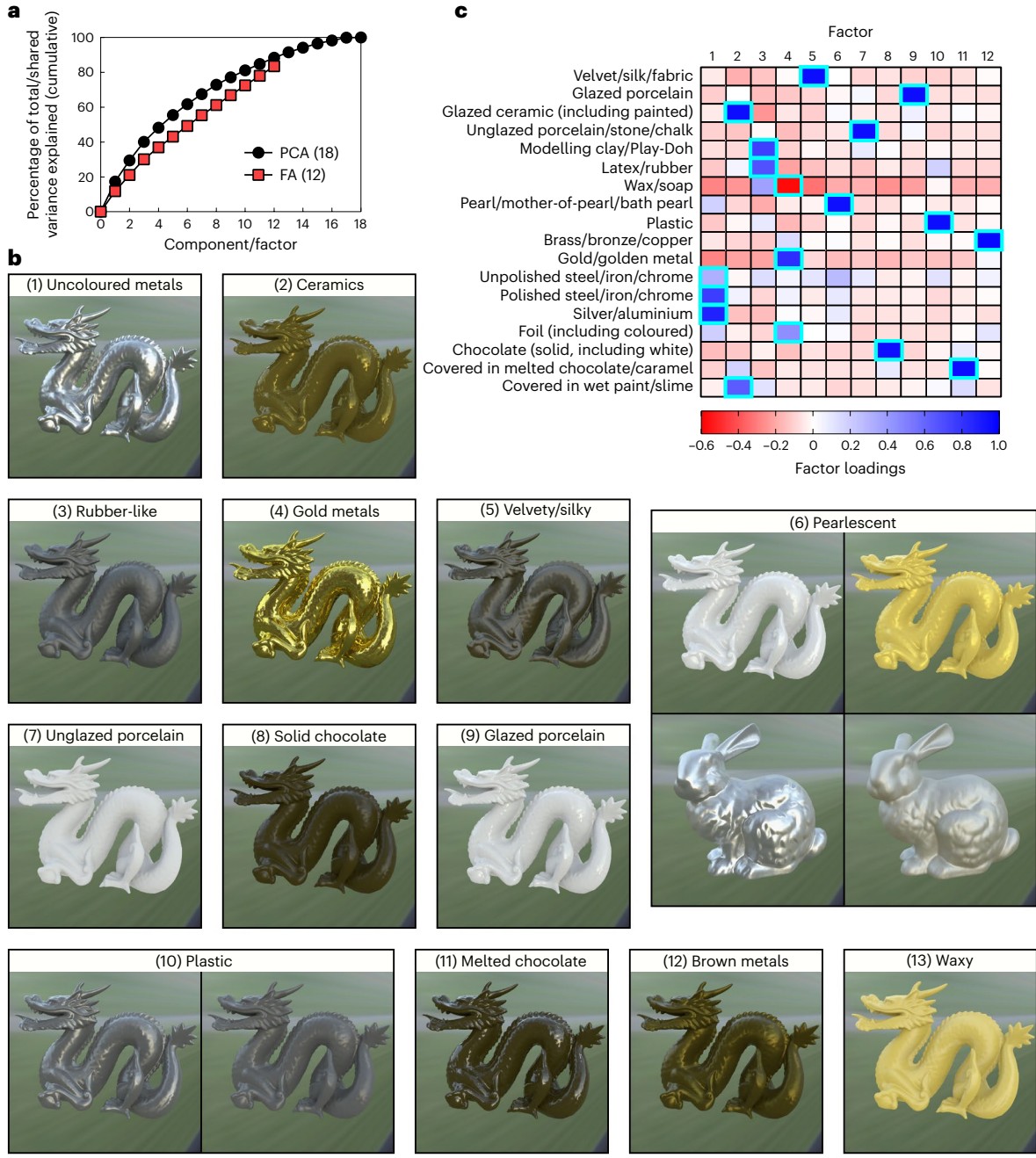

**Fig. 4 | Factor analysis performed on stimulus profiles obtained from the 18-AFC task. a**, Black circles indicate the cumulative total variance explained by each component from an initial PCA. Red squares are the cumulative shared variance explained by each factor from the 12-factor solution, which accounted for 80% of the shared variance between stimulus profiles. **b**, Example stimuli from the 13 emergent material dimensions from the factor analysis (FA). Supplementary Fig. 8 shows more example stimuli for each shape and light field. **c**, Heat plot of the factor loadings for each category from the 18-AFC task, with blue and red cells showing positive and negative loadings, respectively. The highlighted cells show the factor onto which each category loaded most strongly. Note that the 13th dimension emerged from the negative loadings onto factor 4. The emergent dimensions were highly interpretable from the category loadings onto the factors; for example, uncoloured metals like steel and silver all loaded strongly onto factor 1, forming a single dimension. We gave material category labels to each dimension (shown in **b**); however, note that these are arbitrary and are only included for interpretative convenience. Supplementary Fig. 7 shows the results of a PCA that retained all dimensions in addition to the results of other factor solutions, whose dimensions overlap with those here.

relates to how bright the specular highlights look in relation to the surrounding regions (Fig. 2c).

We found that intersubject agreement for gloss ratings was high (median $r = 0.70$), and overall a linear combination of the gloss cues (coverage, sharpness and contrast) accounted for 76% of the variance in participant's gloss ratings (Fig. 5a); $R^2 = 0.76$, $F(3,916) = 973.74$, $P < 0.001$. This links perceived gloss with objective image-based measures of specular reflection features for a wide range of reflectance conditions. However, the material dimensions defined in Experiment 2 were not associated with a particular level of gloss, contrary to what would be predicted by the feedforward hypothesis (Fig. 1b, top). Instead, stimuli from the same material class exhibited a wide distribution of gloss levels, and stimuli from very visually distinct classes like ceramic and (gold and uncoloured) metals had completely overlapping

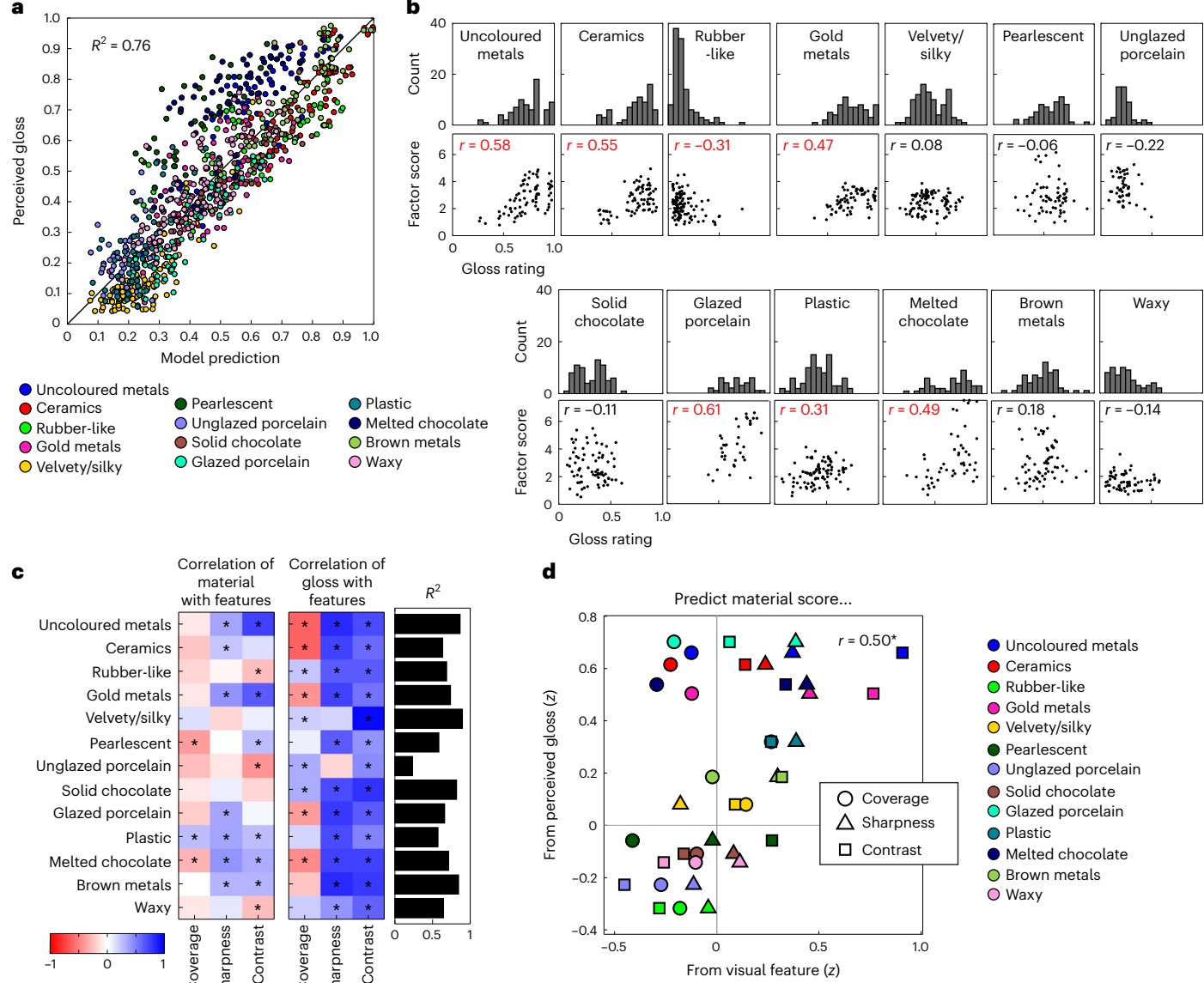

**Fig. 5 | Material class is not determined by but may constrain gloss perception. a**, Results of a linear regression model predicting perceived gloss from coverage, sharpness and contrast (gloss cues). The model accounts for 76% of the variance in gloss ratings between stimuli and is comparable with a model that predicts participant's gloss ratings from each other (leave one out; average $R^2 = 0.72$). **b**, Histograms show that stimuli from the same material class exhibited a wide distribution of gloss levels, which persisted even when only the strongest loading stimuli on each dimension were considered (Supplementary Fig. 10). Although some dimensions have a narrower range of gloss levels than others, stimuli from very visually distinct material dimensions like ceramics and (gold and uncoloured) metals have completely overlapping distributions of gloss ratings. Subjacent scatter plots show correlations between perceived gloss and material score, for each material class. Red coefficients indicate statistically significant correlations (uncoloured metals: $r = 0.58$, $P < 0.001$;

ceramics: $r = 0.55$, $P < 0.001$; rubber-like: $r = -0.31$, $P = 0.001$; gold metals: $r = 0.47$, $P < 0.001$; glazed porcelain: $r = 0.61$, $P < 0.001$; plastic: $r = 0.31$ $P = 0.004$; melted chocolate: $r = 0.49$, $P < 0.001$). Black coefficients indicate correlations are not statistically significant (all $P > 0.05$). The results hold for other factor solutions (Supplementary Fig. 11). **c**, The variance in gloss accounted for by the gloss cues differed within each material class ($R^2$, black bars), and the strength and direction of each cue's correlation with gloss ratings/material score differed across materials (Pearson correlation colour coded according to strength and direction; asterisked correlations indicate $P < 0.05$; see Supplementary Fig. 13 for correlation between cues). **d**, The correlations between perceived gloss and material score (from **b**) are confounded with the material scores correlating with the gloss cues themselves. In **d**, $z$ stands for Fisher-transformed correlation coefficients (Pearson correlation). Data points in **a** and **d** are colour coded by material category (see legend for details).

gloss distributions (histograms in Fig. 5b). We investigated whether the wide gloss distributions could be caused by the continuous nature of the material dimensions (for example, perhaps more metallic-looking materials are glossier, more rubber-looking materials are less glossy, and so forth). If this is true, then gloss ratings should correlate with the loading of stimuli onto material dimensions (material factor scores). This was the case for only seven of the 13 material dimensions (scatter

plots in Fig. 5b, correlation coefficients highlighted in red), with the other six showing no significant correlation between gloss ratings and material score (black coefficients), suggesting that gloss level is overall not a good indicator of how well stimuli load onto a material dimension.

The lack of a clear relationship between perceived gloss and material raises the possibility of a direct link between visual features and perceived material. Indeed, we found that material (factor) scores

correlated directly with one or more of the cues for gloss (that is, the visual features coverage, sharpness and contrast) for 11 of the 13 material dimensions (Fig. 5c, left). This relationship between cues and material scores can explain the extent to which gloss ratings predicted material scores in Fig. 5b (shown in Fig. 5d, $r = 0.50$, $P = 0.001$), suggesting that an object's material could be computed from features of specular image structure directly rather than via gloss estimation. Supplementary Analysis A supports this idea by showing that a model predicting material score from gloss level performed worse than a model predicting material score directly from a linear combination of the cues (Supplementary Fig. 12). Moreover, the predictiveness of cues to perceived gloss varied across different material classes (black bars in Fig. 5c), suggesting that, rather than gloss mediating cues to material, the visual cues used to estimate gloss could instead be constrained by perceived category. In the Discussion we explore the possibility that surface gloss and material could be co-computed and mutually constrained by the same image structure (Supplementary Analysis A). Collectively, these results do not support the idea that gloss mediates the perception of materials from specular reflection image features (feedforward hypothesis; Fig. 1b, top), but are in line with the idea that surface gloss and material class could be simultaneously 'read out' from these features (simultaneous hypothesis; Fig. 1b, bottom).

## Features of specular image structure predict material class

If visual features directly determine the perceived material of an object, this probably includes colour information. Indeed, many material dimensions that emerged from our stimulus set seem to be defined by specific colour information produced by reflectance properties, including gold, brown and uncoloured metals; melted and solid chocolate; glazed and unglazed porcelain; and waxy materials (Supplementary Fig. 8). We measured three visual features—highlight saturation, lowlight saturation and lowlight value—that capture colour information within and outside the specular highlight regions (Fig. 2d–f). These features, or colour cues, are linked to different aspects of a material's colour appearance (for example, surface pigment, or specular 'tint' seen in coloured metals). Intuitively, highlight saturation is the colour saturation within the specular highlight regions (bright spots); lowlight saturation is the colour saturation outside the highlight regions (referred to as lowlight regions[25]); and lowlight value is the brightness of the colour within the lowlight regions.

To test the prediction that materials can be discriminated directly from features of specular image structure (simultaneous hypothesis; Fig. 1b, bottom), we sought to predict material class from the measured visual features (the three gloss cues and three colour cues) (Fig. 2) for the stimuli that loaded most strongly onto each material dimension (Fig. 4). Figure 6a plots the distribution of visual features for these stimuli (coloured violin plots) for each material. Visualized in this way, the features provide 'material signatures' for each class. The data were subjected to a linear discriminant analysis (LDA) that classified materials based on linear combinations of features. We used a leave one condition out approach, in which the classifier was trained on three of the four shape/lighting conditions (for example, dragon–kitchen, bunny–kitchen, dragon–campus) and tested on the remaining

condition (for example, bunny–campus). Figure 6b plots the accuracy of the model for each material, combined over the four training–test combinations. Overall accuracy was 65%, which is well above chance (7.7%, red dotted line). A further cross-validation test showed that the model generalized across shape and lighting conditions (Fig. 6c, mean accuracy 64%, s.d. = 5.2), demonstrating that features of specular structure can predict human material categorization behaviour for our stimulus set. Figure 6d,e illustrates that the discriminations made by the model are perceptually intuitive.

Interestingly, some materials were classified better than others (Fig. 6b). The materials with the highest classification accuracies (Fig. 6b) were those with the most distinct material signatures (Fig. 6a; waxy materials 100%, uncoloured metals 95%, unglazed porcelain 94%, solid chocolate 89%, gold metals 86%, glazed porcelain 72%, brown metals 69%, melted chocolate 61%). That is, there are at least a few features that seem to 'characterize' those materials (for example, uncoloured metals must have uncoloured highlights and lowlights; waxy materials must be light and coloured with low-contrast reflections). Materials with the lowest classification accuracies had a less distinct set of features (ceramics 36%, pearlescent materials 31%, plastic 28%, velvety/silky materials 28%). One possible reason for this is that there are, for example, many types of plastic or pearlescent material (Fig. 4b), and different specific (nonlinear) combinations of features define these subtypes—something that LDA does not capture. A second possibility is that the stimuli might not fall nicely into perceptually discrete classes and would be better represented as a smooth continuation from one material dimension to another. To investigate this second possibility, we tested whether variations in category profile between stimuli (Fig. 3b) could be predicted by variations in the measured visual features (Fig. 2). Using representational similarity analysis[33] we found that the six visual features predict perceived differences in material between stimuli and account for more variance than when other predictors (reflectance parameters, or gloss ratings combined with colour cues) are used (Supplementary Analysis B and Supplementary Fig. 15).

## Manipulating specular structure transforms material class

Thus far, our analyses have been correlational in nature. If material class is computed from combinations of the visual features that we measured (Fig. 2), then directly manipulating these features should transform perceived category in predictable ways. This would better test whether the measured features are causally responsible for the perceived category shifts, or whether they merely correlate with other changes in image structure that are important for material perception that we did not measure.

To this end, we attempted to directly manipulate visual features to transform stimuli that were perceived as glazed ceramic in Experiment 2 into each of the remaining materials, for each shape and light field (Fig. 7a). We created two stimulus sets to test the success of our feature manipulations. The first set contained 'simple' (linear) feature manipulations, which correspond closely to the previously measured visual features (Supplementary Fig. 16). The second set contained 'complex' (nonlinear) feature manipulations, which accounted for particular types of contrast needed for some materials that we empirically

**Fig. 6 | LDA predicting material category from visual features. a**, Radial violin plots show the distribution of measured visual features for each material. For each feature, solid lines correspond to the 50th percentile and dashed lines to the 25th and 75th percentiles of the distribution ($n = 36$ stimuli per plot). See text for details. Hi. satu., highlight saturation; Lo. satu., lowlight saturation; Lo. val., lowlight value; Sharp., sharpness. **b**, The results of a linear classifier with a leave one condition out validation procedure. The red dotted line indicates chance level (1/13). **c**, Classification accuracy generalizes across different illumination and shape conditions. See Supplementary Fig. 14 for similar results with other dimension reduction solutions. **d**, The stimuli are plotted in linear discriminant space (LD1 and LD2 are linear discriminant 1 and 2, respectively).

Points are colour coded by either category (left), or visual features (right). **e**, The same stimuli are plotted for different linear discriminants (top, LD1 versus LD4; bottom, LD2 versus LD3). These plots illustrate that the discriminations made by the model are perceptually intuitive. For example, a combination of average saturation and brightness within the lowlight region can be used to discriminate materials like porcelain (bright, uncoloured body) and waxy materials (bright, coloured body) from other materials with darker body colours; saturation within the highlight region is useful for discriminating brown metals (coloured highlights) from solid chocolate (uncoloured highlights). See Supplementary Table 1 for LDA weights.

noticed were not captured by the first set of manipulations (Supplementary Fig. 16). For example, the velvety/silky stimuli from Experiment 2 were defined exclusively by very rough, anisotropic specular reflections, which caused a high degree of directional blur. This gives

the effect of elongated, low-clarity specular reflections that, despite this low clarity, have rapidly changing (sharp/high-contrast) boundaries between highlights and lowlights relative to isotropic surfaces (Figs. 3a and 4b and Supplementary Fig. 8). On the other hand, the

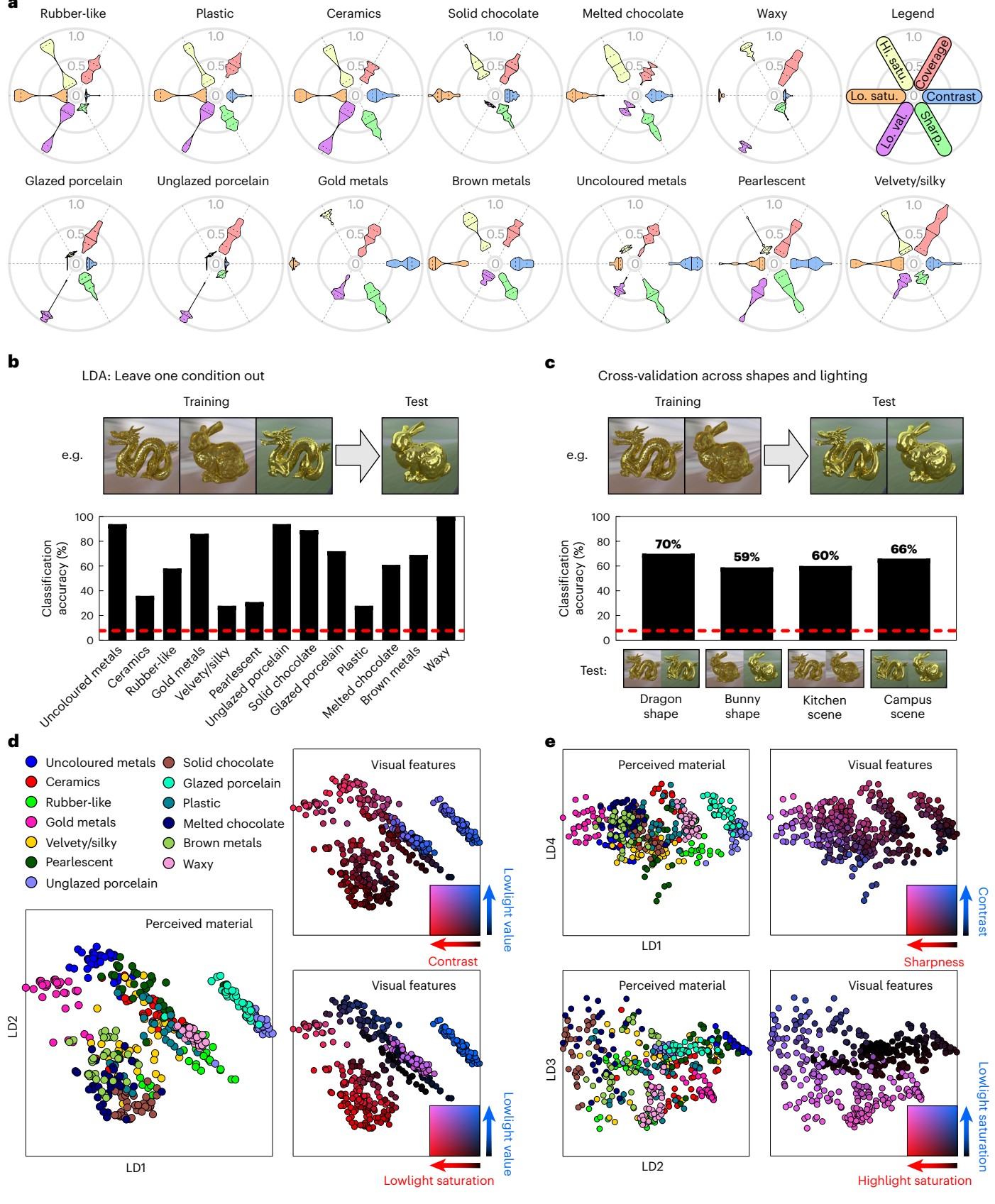

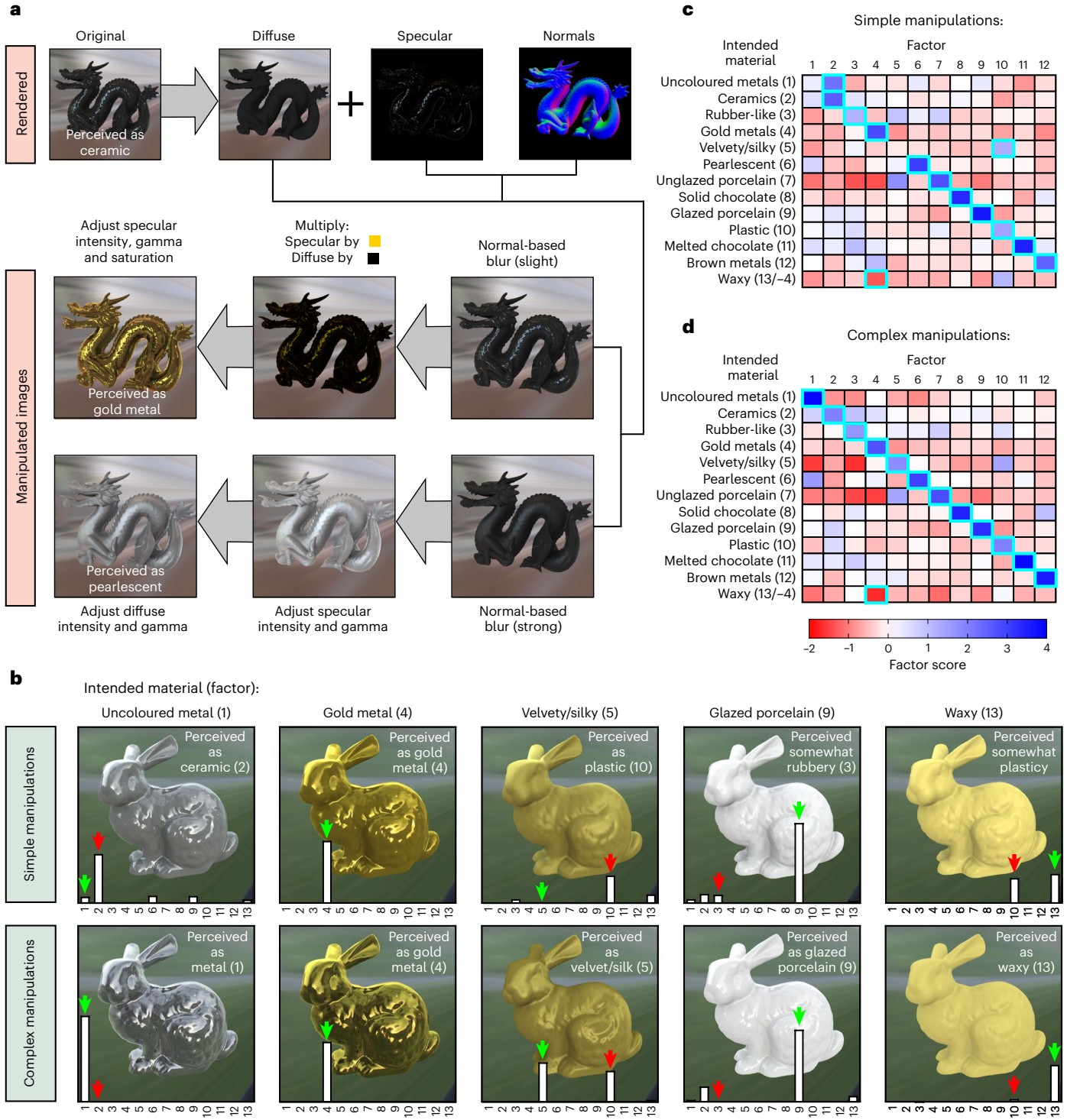

**Fig. 7 | Manipulating visual features transforms perceived material category. a**, Manipulations were performed on specular and diffuse images separately before being recombined: as with the original feature measurements, the specular component was obtained by subtraction of the diffuse component from the full rendering (Supplementary Fig. 9). The sharpness of reflections was first modified by blurring pixels of the specular component that have sufficiently similar normals (Methods). The subsequent filters adjusted the colour and intensity of each component. For example, for materials like gold (top manipulated image), the specular component is first multiplied by a colour (the diffuse component is multiplied by 0), then its intensity and saturation are adjusted. For materials like pearl (bottom manipulated image), the intensity of the diffuse component is also adjusted. See the Methods for a detailed

description of each filter, along with the special filter used for velvet, based on a non-monotonic remapping of the specular component. **b**, Example stimuli after simple manipulations (top) and complex manipulations (bottom). White bars show the stimulus factor scores after participant judgements (main text). After applying simple feature manipulations, materials sometimes resembled unintended categories (red arrows). The perceived materials align much better with the intended category (green arrows) after the complex feature manipulations. **c,d**, Heat maps showing average factor scores for stimuli from each intended category after simple (**c**) and complex (**d**) feature manipulations, respectively. The highlighted cells show the factor onto which each category most strongly loaded.

perception of uncoloured metals appears to require the presence of high-contrast reflections that appear all over the surface (that is, there is a high-contrast image structure in the lowlights, not just the highlights; see also Norman et al.[13]). These particular types of contrast could not be achieved through a simple linear transformation of image intensities from the glazed ceramic stimuli but were accounted for after applying more complex additional filters to nonlinearly manipulate pixel intensities (Fig. 7b and Supplementary Fig. 17).

In Experiment 4, a new set of participants (*n* = 22) performed an 18-AFC task identical to Experiment 2, but with the feature-transformed stimuli. The stimulus profiles were converted to factor scores to see which material dimensions best applied to the new stimuli. The average factor scores plotted in Fig. 7c,d show that participants agreed with our informal observations. For the first stimulus set (linear feature manipulations; Fig. 7c), stimuli that were intended to be uncoloured metals and velvety/silky materials actually had the quality of glazed ceramic and plastic, respectively. For the second stimulus set (nonlinear feature manipulations; Fig. 7d), the perceived materials align much better with the intended classes.

## Discussion

In the present study we showed that the image structure produced by specular reflections not only affects how glossy a surface looks, it can also determine the quality, or category, of the material that is perceived. Critically, specular reflections provide more information than just distinguishing between surfaces that are matte versus glossy[24], or plastic versus metal[16]; we found that the perceptual dimensionality of glossy surfaces defined by a diffuse and specular component is much larger than has been suggested previously[34–37]. This is probably because previous studies on gloss perception have manipulated stimulus parameters within a narrow range and used restricted tasks; for example, stimuli are simple, smooth shapes and/or their reflectance properties encompass only the range of plastic- and ceramic-looking materials, and participants judge the level of gloss or relative similarity between surfaces. Here, we used complex shapes, manipulated reflectance parameters within a wider range and asked participants to judge each object's qualitative material appearance. This greatly expanded the number of dimensions required to account for perceptual differences between glossy surfaces relative to previous studies. Note that we did not sample the whole perceptual space of glossy objects defined by a diffuse and specular component (for example, we manipulated colour saturation within only one hue). The dimensionality of gloss appearance is likely to expand further upon sampling a wider range of reflectance parameters, shapes, mesostructure detail and environment lighting conditions.

Importantly, we found that changes in specular structure—caused by either generative sources or direct feature manipulation—led to qualitative shifts in material appearance beyond those expected by the reflectance function used. This demonstrates that features of specular image structure can be diagnostic for recognizing a wide range of materials. A potential reason for this is that a material's surface reflectance properties create some of its most salient optical characteristics, and, because most surfaces reflect some light specularly, relying on such characteristics could have ecological importance when other cues to material are not available. For instance, the visual effects of translucency can be greatly diminished with frontal lighting[38]; in such cases, visual features caused by specular reflections might remain diagnostic of translucent materials (for example, porcelain). Similarly, mesoscale details like fibres on cloth or scratches on metal might not be visually resolvable when seen at a distance, yet such materials might still be recognized from specular image structure. Indeed, these additional sources of image structure (from mesostructure or translucency) are absent from the stimuli used in the present study, and although such details might render the materials more compelling (for example, when compared with a photograph), the stimuli nevertheless convincingly resemble silk-like, porcelain-like, wax-like, brushed metal-like materials and so on. Interestingly, these results are in line with findings from the computer graphics literature that show that visual effects from different types of fabrics come predominantly from specular reflections, and diffuse reflection and shadowing-masking play a much less pronounced role even for relatively matte-looking fabrics[39].

Our data do not support the notion of a feedforward path to recognition, whereby the visual system combines estimates of physical properties that are first 'recovered' from images; instead, our results are in line with the idea that vision assigns perceptual qualities to statistically varying image structure[2,5,6,40,41]. Specifically, we found that gloss was not a good indicator of a material's class; instead, materials were differentiated directly based on measurable image-based specular reflection features, suggesting that material qualities like gloss are a perceptual outcome with—rather than a component dimension of—our holistic impressions. Indeed, the perception of material qualities like gloss might even be influenced by these holistic impressions, as suggested by the fact that the contribution of different cues to gloss was not stable across different materials, but covaried with the cues that predicted material class. The regions of feature space occupied by different material classes (different qualitative appearances) seemed to mediate the processing of those same features when estimating surface glossiness.

One potential mechanism is that emergent categories from our stimulus set could reflect cognitive decisions, and this cognitive interpretation has a 'top-down' influence on which features are used to judge surface gloss. For example, 'chocolate' and 'plastic' could have similar contrast and sharpness of specular highlights (similar 'gloss types') and the different labels might result from a cognitive decision by participants based on body colour (brown versus yellow). However, we cannot think of a principled reason why a feature's influence on material category would affect how people choose to use that feature for gloss judgements.

Furthermore, we argue that such a clear perceptual scission of image structure into different layers (for example, a gloss layer and a body colour layer) and then subsequent 'cognitive reassembly' is unlikely, especially for the complex-shaped stimuli and wide sampling of reflectance parameters used in the present study. This type of layered appearance of gloss, which can be construed as a form of transparency perception, probably applies only to smooth surfaces with little variation in surface curvature, such as spheres[25]. For our stimuli, specular image structure seems to combine interactively with diffuse shading to create a gestalt-like quality for each material, such that we may not even have perceptual access to individual cues (Supplementary Fig. 1). As with object recognition, this material quality can be given a label like 'gold' or 'pearl', but nonetheless reflects a holistic impression, rather than a cognitive combination of cues (also see Okazawa et al.[42]). Such holistic impressions are probably why we could successfully manipulate visual features to transform one category into another (Experiment 4), but why linear models like LDA and representational similarity analysis (Experiment 2) did not better account for the data.

We propose an alternative mechanism to explain the covariation between cues for surface gloss and material class, inspired by converging evidence from independent lines of psychophysical and neuropsychological research that suggest computations of stimulus properties are inherently coupled[3,43]. Specifically, monkey physiology studies have found that neurons in primate area V4 respond only to specific combinations of texture and shape, and not to these properties separately, suggesting joint representations of shape and surface properties[43]. In line with this, a series of human psychophysics studies have demonstrated that percepts of 3D shape and surface properties (for example, gloss or translucency) are mutually constrained by specific image gradient–contour relationships (that is, the cues for each are not separate)[3]. Similarly, transparency impressions triggered by low-contrast centre–surround displays are mutually constrained with

the perception of surface lightness: a medium-grey patch surrounded by a slightly darker (homogenous) surface can appear whiteish with the quality of a flimsy transparent sheath, which occurs to the extent that luminance within the central patch is attributed to a foreground (patch) or background (surround) layer[44]. In the present study, the covariation between cues for gloss and material class could also reflect a mutual dependency (that is, their computations could be coupled). We suggest that a surface's qualitative (categorical) appearance constrains how specular image structure is perceptually allocated to different aspects of material appearance such as specular reflections (versus say bright pigment, or illumination, or 3D shape[45,46]), and thus which cues are perceptually available to make a surface look 'glossy' or 'shiny'.

Interactions between categorical, or 'holistic', material impressions and the perception of different stimulus properties warrants further investigation because the underlying mechanism will influence how we study perception in multiple domains. Recently, unsupervised deep neural networks have generated excitement as a potential tool to uncover purported mid-level visual processes; for example, by providing evidence that the brain could potentially spontaneously discover the existence of properties like gloss, illumination direction and shape without previous knowledge about the world, but by learning to efficiently represent variations in image structure produced by these properties[47]. However, here we showed that with more complex stimuli, image information cannot be neatly mapped onto perceptual properties like gloss; this mapping is constrained by the qualitative or categorical appearance of surfaces. Some material classes occupy relatively homogeneous feature spaces (for example, gold) compared with other, more diverse classes (for example, plastic), which probably emerges through behavioural relevance. Thus, for unsupervised learning to generate useful hypotheses about perceptual processes, future work needs to consider how the behavioural relevance of stimuli might influence spontaneous learning about statistical image variations.

Neuropsychological results should also be interpreted in the context of qualitative, categorical material perception. For example, neurons in the lower bank of the superior temporal sulcus in monkeys have been shown to respond preferentially to specific combinations of manipulated gloss parameters[48]. It is possible that different categorical or holistic material impressions are triggered by these specific combinations, driving this response. Moreover, research into object recognition[49] often investigates how visual features like texture[50] and form[51] influence object category representations, but thus far have neglected the role of surface properties in recognition; for example, by not controlling for changes in surface properties[51–53], or 'scrambling' the images in a way that breaks the perception of surfaces[50]. Our results suggest that the specific photogeometric constraints on image structure that trigger our perception of surface gloss play an important role in visual categorization and recognition, and that a fruitful direction for neuropsychological research could be to focus on identifying the neural mechanisms that represent objects holistically[54].

However, we think that this role extends beyond object identification. Just as surface gloss as a unitary concept is not a component dimension of material perception, we argue that material perception is not merely a component dimension of scene perception. A material's qualitative appearance is useful for different aspects of navigation, such as identifying where we are (beaches have sand and water), choosing which paths to take (icy versus muddy versus concrete) and deciding which obstacles to avoid (solid rock versus flexible vegetation). We also need material perception for locating items among clutter (finding a metallic pot in the cupboard), evaluating an object's function, usefulness or motor affordances (Can I eat it? Is it expensive or fake? How do I pick it up?) and predicting tactile properties and future states of objects (Is it heavy, sticky or wet? Will shatter, stretch or bounce?). Our results shed light on how image structure is transformed into our representations of surfaces with particular material 'appearances', thereby making an important contribution towards bridging the gap between the early

stages of visual information processing and behavioural goals such as categorization, action and prediction[55,56].

## Methods

### Participants

Volunteer participants were students at Justus Liebig University Giessen in Germany. Fifteen participants completed the free-naming experiment (Experiment 1; mean age 23.7 years; 80% female, 20% male). Eighty participants took part in the 18-AFC experiment (Experiment 2; mean age 24.9 years; 83.3% female, 16.7% male), 22 participants took part in the gloss rating experiment (Experiment 3; mean age 25.3 years; 58.3% female, 41.7% male) and 22 participants took part in the feature manipulation experiment (Experiment 4; mean age 23.4 years; 81.8% female, 18.2% male). One participant was excluded from Experiment 3 because they did not understand the task instructions, resulting in 21 participants. Experiments 2 and 4 included German speaking participants, for which German was their first language. Experiments 1 and 3 included both German and English speaking participants. Different participants were recruited for each experiment.

### Stimuli

**Stimulus generation in Experiment 1 (free-naming), Experiment 2 (18-AFC) and Experiment 3 (gloss ratings).** We generated our stimulus set by computer rendering images of complex glossy objects under natural illumination fields. Each image depicted an object with the illumination field as the background, rendered at a resolution of 1,080 × 1,080 pixels. Object meshes were the Stanford Bunny and Dragon from the Stanford 3D Scanning Repository (Stanford University Computer Graphics Laboratory; http://graphics.stanford.edu/data/3Dscanrep/). Wavefront.obj files of these models were imported into the open-source modelling software Blender (v.2.79) and rendered using the Cycles render engine, which is an unbiased, physically based path-tracing engine.

Stimuli were illuminated by the 'kitchen' and 'campus' light fields from the Debevec Light Probe Image Gallery[57]. Interactions between light and surfaces were modelled using the Principled Bidirectional Scattering Distribution Function (BSDF) shader, which is based on Disney's principled model known as the 'PBR' shader[58]. The Principled BSDF shader approximates physical interactions between illuminants and surfaces with a diffuse and specular component (for dielectric materials), and a microroughness parameter that controls the amount of specular scatter. An advantage of the Principled BSDF shader over other models like the Ward model is that it accounts for the Fresnel effect. The microfacet distribution used is multiple-scattering GGX, which takes multiple bounce (scattering) events between microfacets into account, giving energy-conserving results. Although there are many parameters to adjust in the Principled BSDF shader, we manipulated only the following (details can be found at https://docs.blender.org/manual/en/dev/render/shader_nodes/shader/principled.html):

- Base colour: proportion of light reflected by the R, G and B diffuse components.
- Specular: amount of light reflected by the specular component. The normalized range of this parameter is remapped linearly to the incident specular range 0%–8%, which encompasses most common dielectric materials. Values above 1 (that is, above 8% specular reflectance) are also possible. A value of 5 (the maximum value used) translates to 40% specular reflectance.
- Specular tint: tints the facing specular reflection using the base colour, whereas the glancing reflection remains white.
- Roughness: specifies the microfacet roughness of the surface, controlling the amount of specular scatter.
- Anisotropic: amount of anisotropy for specular reflections. Higher values give elongated highlights along a surface tangent direction given by a tangent field. This field is obtained per

object using the spherical mapping functionality of Blender. Internally, it works in two steps. First, a radial pattern of directions is defined on a 3D sphere enclosing the object, by assigning the direction of the meridian going through each spherical point. Such a pattern has singularities at sphere poles, where all directions converge. Second, each point $p$ on the object is mapped to a point $q$ on the enclosing sphere by tracing a ray from the object centre through $p$. The direction at $q$ is then projected on the local tangent plane at $p$ and renormalized to yield the local tangent. Note that singularities remain singularities after projection.

- Anisotropic rotation: rotates the direction of anisotropy in the local tangent plane, with a value of 1.0 indicating a rotation of 360°.

**Rendering parameters in Experiment 1 (free-naming).** A total of 270 stimuli were generated for the free-naming experiment. Stimuli were generated by rendering combinations of the parameters for a dragon model embedded in the kitchen light field:

- Base colour: six diffuse base colours based on two levels of saturation (greyscale, saturated yellow) and three levels of value (dark, medium and light). For the greyscale stimuli, the red, green and blue (RGB) values were equal for each lightness level (RGB = 0.01, 0.1 and 0.3 for dark, medium and light stimuli, respectively). The RGB values for saturated stimuli were [0.01, 0.007, 0.001] for dark stimuli, [0.1, 0.074, 0.01] for medium stimuli and [0.3, 0.221, 0.03] for light stimuli.
- Specular: three specular levels: 0.1, 0.3 and 1, which correspond to 0.8%, 2.4% and 8% specular reflectance, respectively.
- Specular tint: two specular tint levels for saturated stimuli: 0 (no specular tint) and 1 (full specular tint).
- Roughness: four roughness levels: 0, 0.1, 0.3 and 0.6.
- Anisotropic: two anisotropic levels for stimuli with roughness levels greater than zero: 0 and 0.9.
- Anisotropic rotation: two anisotropic rotations for anisotropic stimuli: 0 (no rotation) and 0.25 (90° rotation).

**Rendering parameters in Experiment 2 (18-AFC) and Experiment 3 (gloss ratings).** A total of 924 stimuli were generated for the 18-AFC and gloss ratings experiments. Stimuli were generated by rendering combinations of the parameters for two shapes (dragon, bunny) and two light fields (kitchen, campus):

- Base colour: six diffuse base colours based on two levels of saturation (greyscale, saturated yellow) and three levels of value (dark, medium and light). For the greyscale stimuli, RGB values were equal for each lightness level (RGB = 0.01, 0.03 and 0.3 for dark, medium and light stimuli, respectively). The RGB values for saturated stimuli were [0.01, 0.007, 0.001] for dark stimuli, [0.03, 0.022, 0.003] for medium stimuli and [0.3, 0.221, 0.03] for light stimuli.
- Specular: four specular levels for the darkest four diffuse shading levels: 0.1, 0.3, 1 and 5, which correspond to 0.8%, 2.4%, 8% and 40% specular reflectance, respectively. The light-coloured stimuli were only rendered with the first three specular levels.
- Specular tint: two specular tint levels for saturated stimuli: 0 (no specular tint) and 1 (full specular tint).
- Roughness: three roughness levels: 0, 0.3 and 0.6.
- Anisotropic: two anisotropic levels for stimuli with roughness levels greater than zero: 0 and 0.9.
- Anisotropic rotation: two anisotropic rotations for anisotropic stimuli: 0 and 0.25.

**Stimulus generation in Experiment 4 (feature manipulations).** We generated the feature-manipulated stimulus set with an approach similar to the compositing techniques used in visual effects. For our manipulation we need several image components of an input scene: a full rendered image, an image of the object rendered with diffuse shading only (diffuse component), a binary mask image and an image of the surface normals. All of these components were rendered with the Cycles engine in Blender. A manipulated image is obtained via the sequence of steps depicted in Fig. 7a: a specular component is first obtained by subtracting the diffuse component from the full rendered image; the sharpness of this specular component is reduced via a normal-based blur operator; the diffuse and specular components are then optionally multiplied by a colour; finally, the intensity and saturation of the resulting specular and diffuse components are adjusted and the final image is obtained by addition of these components (specular + diffuse). These steps are explained in more detail below.

The normal-based blur operator is implemented with a cross bilateral filter[59] on the specular component and takes into account the orientation of the surface normals. The cross bilateral filter consists of an image convolution with a modified blur kernel: it not only weights the contribution of neighbour pixel colours given their distance (the spatial weight), but also based on additional pixel-wise information (the range weight). The two weights are combined multiplicatively, and the blur kernel is systematically normalized so that the sum of combined weights is one. In our case, the space weight is given by a box kernel of a large static size (160 × 160 pixels), which merely serves to limit the set of pixels that are considered for blurring. The range weight is given by an exponentiated normal dot product: $(\mathbf{n} \cdot \mathbf{n_0})^s$, where $\mathbf{n_0}$ is the normal at the centre pixel in the kernel, $\mathbf{n}$ is the normal at a neighbour pixel and $s$ is a shininess parameter. This simple range weighting formula is directly inspired by Phong shading: large shininess values result in a sharp angular filter, whereas small shininess values result in a broad angular filter, which has the effect of blurring specular reflections. In practice, we use a blur parameter $b$ in the [0,1] range, which is empirically remapped to shininess using $s = 1 + 1/((b/5)^4 \times 100 + \varepsilon)$, where $\varepsilon$ is a small constant used to avoid division by 0.

Some particular categories of materials such as gold or chocolate require a colourized diffuse or specular component. This is obtained by simply multiplying the image component by a colour: sCol for specular, dCol for diffuse. In some cases (gold and silver), the diffuse component is discarded entirely, which is equivalent to having dCol = (0,0,0), as indicated in Fig. 7a. We chose to use bright and saturated colours in all other cases to make the parameters in the next step more easily comparable. For the last intensity and colour adjustment step, each image component is first converted to the hue, saturation and value (HSV) colour space. The following manipulations apply either to the specular or diffuse component, with the corresponding parameters prefixed by either 's' or 'd'. The value channel is manipulated by a combination of gamma exponentiation (controlled by dGamma and sGamma) and multiplication (by sBoost or dBoost) using the following simple formula: Boost × $v^{Gamma}$, where $v$ is value in the HSV colour space. The saturation channel is also multiplied (by sSat or dSat). Both components are converted back to RGB and added together to yield the final image.

We use an additional filter for the specific case of velvet/satin. It is applied to the specular component right after normal-based blurring. The specular component is multiplied by a first mask $m_1$ that is computed via nonlinear mapping of the luminance $l$ of the specular component. We use the following formula: $m_1 = f(0, l_m/2, l)$ if $l < l_m/2$, and $m_1 = 1 - f(l_m/2, l_m, l)$ otherwise, where $l_m$ is the maximum luminance in the specular component image and $f(a,b,x)$ is the 'smooth step' function defined in the OpenGL Shading Language that maps $a \le x \le b$ to the [0,1] range (see https://www.khronos.org/registry/OpenGL-Refpages/gl4/html/smoothstep.xhtml for details). The effect of multiplication by $m_1$ is to darken the core of the brightest highlights, hence producing elongated highlight loops as seen in Supplementary Figs. 17 and 18. It is similar in spirit to the sinusoidal modulation of Sawayama and Nishida[60] (Figure 11 in their paper), except that it is only applied to the specular component and with a different nonlinear remapping.

We have also found it necessary to multiply the specular component by a second mask $m_2$ to slightly attenuate some of the elongated highlights. This mask is computed using a simple normal dot product: $m_2 = (\mathbf{n} \cdot \mathbf{n}_m)$, where $\mathbf{n}$ is the normal at a pixel and $\mathbf{n}_m$ is a manually set direction that is chosen per scene, pointing to the direction of the main highlight. This has the effect of mimicking the unconventional darkening observed in images of velvet materials at locations where the main highlight should occur.

**Manipulation parameters in Experiment 4 (feature manipulations).** For each of the four scenes showing either of two objects (bunny and dragon) in either of two lighting environments (campus and kitchen), we have taken as input a material configuration previously classified as 'glazed ceramic', from which we have produced 12 manipulations for each of the other material categories. There were also two manipulation conditions (below), yielding a total of 104 stimuli for the feature manipulation experiment.

The input material was systematically chosen to have a grayscale base colour of 0.1, a specular level of 0.3, a specular tint of 0, a roughness of 0 and an anisotropic level of 0.

The manipulation parameters are listed in Supplementary Tables 2 and 3. In the 'simple' manipulation condition (Supplementary Table 2), all the Gamma (nonlinear) parameters were set to 1 and the special velvet filter was discarded. In the 'complex' manipulation condition (Supplementary Table 3), setting some of the sGamma and dGamma parameters away from 1 had an impact on the intensity of either the specular or diffuse component; as a result, we also had to adjust the sBoost and dBoost parameters. We applied the velvet filter only for the velvet manipulation in this condition, using the following $\mathbf{n}_m$ directions: (0.57, −0.53, 0.63) for bunny/campus, (−0.22, −0.48, 0.85) for bunny/kitchen, (−0.62, 0.7, 0.35) for dragon/campus and (−0.72, 0.57, 0.39) for dragon/kitchen.

## Procedure

**Stimulus presentation.** Stimulus presentation and data collection were controlled by a MATLAB script (release 2018b, Mathworks) using the Psychophysics Toolbox v.3 (ref. 61). In the free-naming task, stimuli were projected onto a white wall in a classroom. Thick black cloth was used to block light from the windows, so that the only source of light came from the projected image. For all other experiments the stimuli were presented on a Sony OLED monitor running at a refresh rate of 120 Hz with a resolution of 1,920 × 1,080 pixels controlled by a Dell computer running Windows 10. Stimuli were viewed in a dark room at a viewing distance of approximately 60 cm. The only source of light was the monitor that displayed the stimuli.

**Task. Experiment 1 (free-naming) task.** Fifteen participants gathered in a classroom and completed the task at the same time. Two authors, A.S.C. and K.D., were present in the room with the participants. This was a qualitative data collection session, and no specific hypothesis was considered at this stage. Participants viewed stimuli one at a time and were asked to classify the material of each stimulus, with no restrictions. They were provided with sheets of paper with space for each trial number to write down their answers. The experimenter controlled the stimulus presentation with a keyboard press. Each trial was presented for as long as it took for all participants to finish writing down their responses. A blank screen was shown for 1 second between each trial. The experiment took approximately 3 hours to complete, including breaks.

The instructions, written on a sheet of paper in both English and German, were as follows:

You will be shown 270 images, and your task is to write down your impressions of the material of each object; that is, what does it look like it is made of? Below are some suggestions of materials that might help prompt you. Your answers might not include, nor are they limited to,

the suggestions below. There are no right or wrong answers and you can respond however you like. You can be as specific (for example, aluminium foil, polyethylene) or general (for example, metal, plastic) as you like. You can also write down more than one material (for example, 'looks most like glazed ceramic but could also be plastic'), or even say that it doesn't look like anything you know. If you can't remember the name of a material, you can write for example 'the stuff that X is made out of'.

The following examples were provided below the instructions:

- Metal; for example, silver, gold, steel, iron, aluminium, chrome, foil
- Textiles; for example, velvet, silk, leather
- Plastic; for example, polyvinyl chloride (PVC), nylon, acrylic, polyethylene, Styrofoam
- Ceramic; for example, porcelain, china
- Minerals; for example, stone, concrete, rock
- Coatings; for example, glazed, painted, enamel, polytetrafluoroethylene (PTFE)/Teflon
- Other; soap, wax, chalk, pearl, composite materials.

**Experiment 2 (18-AFC) task.** The 924 stimuli were randomly split into four sessions, so that each participant categorized a quarter of the stimuli (20 participants per stimulus). The experiment was self-paced with no time constraints, and for most participants the experiment lasted approximately 1–1.5 hours. Only experimenter and participant were present during the experiment.

In each trial, observers were presented with an object in the centre of the screen (29° visual angle) with 18 categories displayed along the sides of the stimulus (Supplementary Fig. 5). They were asked to choose the category that best applied to that stimulus. If they were unhappy with their choice, they could change the confidence rating at the bottom right of the screen. Before the experiment, participants were asked to read carefully through the list of materials and were shown several examples of the stimuli to be presented. The purpose of this was so observers got a sense of the range of materials in the experiment. Observers were restricted to choosing only one category for each stimulus, and were given the following instructions verbally by the experimenter:

Use the mouse to click on the category that best describes the material of each object. When you are satisfied with your choice press the space bar to proceed to the next trial. In the case of categories with multiple items (for example, velvet/silk/fabric), the perceived material only needs to apply to one, not all, the categories. There are no right or wrong answers, as the experiment is about the perception of materials. Not all categories will have an equal number of stimuli—you may choose one category more or less than others (or not at all). If you are not satisfied or confident with your choice, change the confidence rating at the bottom of the screen. This should be used in the case you feel that none of the available categories fit; if you think more than one category applies then just choose the most suitable option.

**Experiment 3 (gloss ratings) task.** The experiment was self-paced with no time constraints, and for most participants the experiment lasted approximately 1 hour. Only experimenter and participant were present during the experiment.

Before the experiment, participants were shown real-world examples of glossy objects (some of which are shown in Fig. 1a). As they were shown these images, they were given the following instructions:

Many objects and materials in our environment are glossy or shiny. Glossy things have specular reflections, and different materials look glossier/shinier than others. You will be shown different objects and asked to rate how glossy each object looks. This experiment is about visual perception so there is no right or wrong answer—just rate how glossy each object looks to you.

Participants were shown many example trials before starting the experiment so that they could experience the full range of stimuli and calibrate their ratings to the range of gloss levels in the experiment. In each trial, observers were presented with an object in the centre of the screen (29° visual angle). The task was to rate how glossy the object was by moving the mouse vertically to adjust the level of a bar on the right side of the screen. They were told to look at many points on the object before making their decision. The starting level of the rating bar was randomly set on each trial.

**Experiment 4 (feature manipulations) task.** Each participant categorized all 104 stimuli, and the task was identical to the 18-AFC task in Experiment 2. The experiment was self-paced with no time constraints, and for most participants the experiment lasted approximately 45 minutes. Only experimenter and participant were present during the experiment.

## Analyses

### Reduced set of category terms. Terms retained in Experiment 1 (free-naming).

A student assistant went through participants' responses for the free-naming task and extracted all category terms (nouns) and descriptors (for example, adjectives describing fillings, coatings, finishes and states), translating them into English. Each participant often used multiple terms per stimulus, all of which were recorded as separate entries. Similar terms (like foil/metal foil/aluminium foil, or pearl/pearlescent/mother of pearl/bath pearl) were combined into a single category. Some terms like 'Kunststoff' and 'Plastik' were considered duplicates because they translated to a single term (plastic) in English.

### Terms retained in Experiment 2 (18-AFC).

We decided to retain the category terms from the free-naming task that at least 5 of 15 participants used. This was an arbitrary cut-off with the aim of being quite inclusive while at the same time maintaining some consensus among participants. Supplementary Fig. 2 shows the 29 category terms that survived this cut-off. Subsequently, two of the authors (A.C.S. and K.D.) worked together to reduce this set of category terms. For example, materials that were visually or semantically similar were combined (specifically, fabrics, including velvet and silk; unglazed porcelain, stone and chalk; latex and rubber; wax and soap; brass, bronze and copper; iron and chrome; and silver and aluminium). To have some numerical guidance for this reduction we computed the correlation between each pair of categories, calculated from the number of participants that used each term for each stimulus (Supplementary Fig. 4). That is, for each pair of categories, correlated vectors were indexed by stimulus number with the values being number of observer responses. The superordinate category 'metal' was not included separately. In the free-naming task, non-category terms were often required to distinguish particular coatings (glazed/varnished), finishes (polished) or states (liquid/melting) (Supplementary Fig. 2). For the 18-AFC task, we separated categories where these terms applied; for example, we included both glazed porcelain and unglazed porcelain, both liquid and solid chocolate, and also included a 'covered in wet paint' category. Thus, the final set of 18 category terms (Supplementary Fig. 5) that participants could choose from in the 18-AFC task was as inclusive as possible while not being too large to overwhelm participants.

### Factor analysis in Experiment 2 (18-AFC).

The correlations between different categories in terms of their stimulus profiles (Supplementary Fig. 6) indicate that some category terms were not independent. This suggests the existence of underlying common dimensions; that is, participants used the same underlying criteria for different category terms. An exploratory factor analysis was performed on the stimulus profiles from Fig. 3b, which allowed us to explain some of this covariation and reveal the underlying category space of our stimulus set. Fig. 4a

(red squares) shows that there is a steady increase in the common variance explained by each additional factor. The amount of additional variance explained by each factor did not drop off after a certain number of factors, indicated by the absence of a plateau in this plot. Therefore, we extracted 12 factors (the upper limit based on degrees of freedom), which were interpretable and explained more than 80% of the common variance between stimulus profiles. This is similar to or greater than the amount of variance explained by the factors/components retained in other material perception studies that have used factor analysis or principal components analysis (PCA)[62]. The factors were interpretable (Fig. 4b) and were labelled based on the original categories that loaded most strongly onto each factor (Fig. 4c). Factor scores were calculated using a weighted least-squares estimate (also known as the 'Bartlett' method). For comparison, Supplementary Fig. 7 shows the results of a PCA that retained all dimensions in addition to the results of other factor solutions, whose dimensions overlap with those here.

### Visual features. Visual feature measurements in Experiment 2 (18-AFC) and Experiment 3 (gloss ratings).

Calculations for most of the visual features relied on specular highlight coverage maps. For glossy surfaces with uniform reflectance properties (like the stimuli used in the present study), specular reflections cover the entire surface. However, for low-gloss objects we often only see specular highlights, or 'bright' reflections, which are usually reflections of direct light sources like the sun, or a lamp. For very shiny surfaces (like metal) and in some lighting conditions we also see lowlights, or 'dark' reflections[25], which are reflections of indirect light coming from other surfaces in the scene. We chose to define coverage as the amount of the surface covered in specular highlights, excluding lowlights, which is consistent with previous literature[32,63]. Marlow and colleagues measured the coverage, contrast and sharpness of specular highlights using perceptual judgements from participants, owing to the difficulty in segmenting the specular component of an image from other sources of image structure (such as diffuse reflectance and shading). An acknowledged concern of this approach is that it uses one perceptual output (perceived coverage, contrast and sharpness) to match another (perceived gloss). It is unclear what image structure observers use to judge each feature, and participants might conflate their judgements of the visual features with each other and with perceived gloss[31] (see also van Assen et al.[64] for a similar use of this method). Therefore, we wanted to develop objective measures of specular reflection features. Currently there is no established method for segmenting specular reflections and diffuse shading from a single image that is robust across different contexts (for example, changes in surface albedo, shape and illumination conditions). To help with this segmentation we rendered additional images that isolated specular and diffuse components.

### Specular reflection segmentation for visual feature measurements.

For each stimulus, a purely specular image was rendered, which had the same specular reflectance as the original (full rendered) image but with the diffuse component turned off. Two purely diffuse images were rendered, which we call 'diffuse image 1' (used to calculate coverage) and 'diffuse image 2' (used to calculate sharpness and contrast; first column in Supplementary Fig. 9). Kim et al.[25] separated specular highlights and lowlights by subtracting an image of a rendered glossy surface (with a specular and diffuse component) from an 'equivalent' fully diffuse image with the same total reflectance as the glossy surface (that is, the surfaces reflected the same amount of light in total; for the diffuse surface light was scattered equally in all directions, and for the glossy surface some light was reflected specularly with the rest scattered diffusely). Kim et al. used the Ward model[65] in which the total reflectance could be easily matched between glossy and diffuse renderings because specular reflectivity is constant at all viewing angles. Because the Principled BSDF simulates the Fresnel effect (whereby specular reflectance increases at grazing viewing angles, depending

on the index of refraction), the diffuse (1) renderings were matched to the facing (along normal) reflectivity of the purely specular renderings. The second diffuse image (diffuse image 2) was created by rendering only the diffuse component of the original (full rendered) stimulus, with the specular component turned off.

Specular reflections were segmented into specular highlights and lowlights by subtracting diffuse image 1 from the purely specular image, which resulted in a subtracted image. This subtracted image was thresholded to serve as a coverage mask (second column in Supplementary Fig. 9). To obtain the best results for our stimulus set, the threshold was set to one-third of the maximum diffuse shading for each stimulus. Pixels above this threshold were considered highlights, and the remaining pixels were considered lowlights. For calculations of sharpness and contrast, specular reflections were segmented from diffuse shading by subtracting diffuse image 2 from the original 'full' stimulus (second column in Supplementary Fig. 9).

**Definition of cues for visual feature measurements.** Visual features (cues) were calculated using RGB images. For gloss cues, coverage was defined as the proportion of object pixels that were calculated to be specular highlights (excluding lowlights); contrast was defined as the sum of root-mean-squared contrast of extracted specular reflections at different spatial frequency bandpasses; the sharpness of extracted specular reflections was calculated for each pixel within the highlight regions using a measure of local phase coherence (we used the MATLAB implementation of the method by Hassen et al.[66], available at https://ece.uwaterloo.ca/~z70wang/research/lpcsi/), and these values were then averaged. For colour cues, highlight saturation and lowlight saturation were calculated as the average colour saturation of pixels within the highlight region, and outside the highlight region (which we call the lowlight region), respectively, as measured by the rgb2hsv function in MATLAB (release 2018b; Mathworks), which converts the red, green and blue values of an RGB image to the hue, saturation and value of an HSV image (https://www.mathworks.com/help/matlab/ref/rgb2hsv.html); lowlight value was calculated as the average value of pixels in the lowlight region (also using the rgb2hsv MATLAB function). A root transformation on sharpness and contrast was applied to linearize the relationship of these cues to gloss ratings.

**Visual feature measurements for Experiment 4 (feature manipulation).** The same visual feature measurements were used for the rendered stimuli (from Experiments 2 and 3) and the manipulated images (from Experiment 4), so that the measurements would be comparable (Supplementary Fig. 16). However, slight modifications had to be made. For the rendered stimuli (Experiment 2), the segmentation between highlights and lowlights (described in the previous section) relied on two additional rendered images: a rendered specular image and a rendered diffuse image (1) with the same total reflectance (Supplementary Fig. 9). For Experiment 4, the image manipulations are all applied to the same 'glazed ceramic' material (Methods). We reused the corresponding rendered specular and diffuse images from the stimulus set from Experiment 2, with one additional step: the specular image was further modified by the normal-based blur filter to account for the change in highlight coverage induced by a change in sharpness. Apart from this, the rest of the visual feature measurement routines remained unchanged.

**Linear discriminant analysis (LDA) in Experiment 2 (18-AFC).** Materials were classified based on linear combinations of visual features using multiclass LDA in MATLAB (release 2018b; Mathworks), which finds a set of linear combinations that maximizes the ratio of between-class scattering to within-class scattering (https://www.mathworks.com/help/stats/discriminant-analysis.html). The MATLAB function *fitdiscr* was used to fit a discriminant analysis classifier to the training data (regularized LDA in which all classes have the same

covariance matrix). To train a classifier, the fitting function estimates the parameters of a Gaussian distribution for each class (https://www.mathworks.com/help/stats/creating-discriminant-analysis-model.html). Test data labels were predicted using the MATLAB function *predict*, in which the trained classifier finds the class with the smallest misclassification cost (https://www.mathworks.com/help/stats/prediction-using-discriminant-analysis-models.html).

**Correlations in Experiment 3 (gloss ratings).** Intersubject correlations for gloss ratings were calculated using Pearson's correlation, and the median was taken. Pearson correlations used for analyses in Fig. 5d,f were Fisher-transformed.

**Linear regression in Experiment 3 (gloss ratings).** A linear regression was performed on 920 stimuli predicting mean gloss ratings from the three gloss cues (coverage, sharpness, contrast). Only stimuli that were allocated a dimension in the factor analysis (from Experiment 2) were included in this analysis (four stimuli loaded negatively onto dimensions, but not enough to make a new dimension for our stimulus set).

### Ethics
This study was approved by the local ethics review board of the Justus Liebig University Giessen (LEK FB 06) and strictly adhered to the ethical guidelines put forward by the Declaration of Helsinki (2013). All participants gave written informed consent before the experiments and were told about the purpose of the experiments. All participants were compensated for their participation at a rate of €8 per hour.

### Reporting summary
Further information on research design is available in the Nature Portfolio Reporting Summary linked to this article.

## Data availability
Psychophysics data and stimuli used in the experiments are available on Zendo: https://doi.org/10.5281/zenodo.5080227. The 3D meshes of the bunny and dragon objects were obtained from the Stanford 3D Scanning Repository and can be found under the following link: http://graphics.stanford.edu/data/3Dscanrep/. Source data are provided with this paper.

## Code availability
Code for analyses (including image analyses) are available on Zendo: https://doi.org/10.5281/zenodo.5080227.

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

## Acknowledgements

A.C.S. and K.D. are supported by a Sofja Kovalevskaja Award endowed by the German Federal Ministry of Education, awarded to K.D. K.D. was also supported by The Adaptive Mind, a research cluster funded by the Hessian Ministry of Higher Education, Research, Science and the Arts, Germany. A.C.S. was also supported by a Walter Benjamin Fellowship funded by the German Research Foundation (DFG). P.B. is supported by l'Agence nationale de la recherche (ANR) project VIDA (ANR-17-CE23-0017). The funders had no role in study design, data collection and analysis, decision to publish or preparation of the manuscript. We thank M. Panayi, C. Baker, F. Schmidt and K. Gegenfurtner for helpful feedback on earlier versions of this manuscript.

## Author contributions

A.C.S. and K.D. contributed to the conception and design of the work; the acquisition, analysis and interpretation of data; and wrote and revised the manuscript. P.B. contributed to the conception and design of the work; the interpretation of data; writing the manuscript and its revisions. P.B. also designed and implemented the image manipulation techniques.

## Funding

## Competing interests

The authors declare no competing interests.

## Additional information

**Correspondence and requests for materials** should be addressed to Alexandra C. Schmid.

# Reporting Summary

Nature Research wishes to improve the reproducibility of the work that we publish. This form provides structure for consistency and transparency in reporting. For further information on Nature Research policies, see Authors & Referees and the Editorial Policy Checklist.

## Statistics

For all statistical analyses, confirm that the following items are present in the figure legend, table legend, main text, or Methods section.

| n/a | Confirmed | |
|---|---|---|
| ☐ | ☒ | The exact sample size (*n*) for each experimental group/condition, given as a discrete number and unit of measurement |
| ☐ | ☒ | A statement on whether measurements were taken from distinct samples or whether the same sample was measured repeatedly |
| ☒ | ☐ | The statistical test(s) used AND whether they are one- or two-sided<br>*Only common tests should be described solely by name; describe more complex techniques in the Methods section.* |
| ☒ | ☐ | A description of all covariates tested |
| ☐ | ☒ | A description of any assumptions or corrections, such as tests of normality and adjustment for multiple comparisons |
| ☐ | ☒ | A full description of the statistical parameters including central tendency (e.g. means) or other basic estimates (e.g. regression coefficient) AND variation (e.g. standard deviation) or associated estimates of uncertainty (e.g. confidence intervals) |
| ☐ | ☒ | For null hypothesis testing, the test statistic (e.g. *F*, *t*, *r*) with confidence intervals, effect sizes, degrees of freedom and *P* value noted<br>*Give P values as exact values whenever suitable.* |
| ☒ | ☐ | For Bayesian analysis, information on the choice of priors and Markov chain Monte Carlo settings |
| ☒ | ☐ | For hierarchical and complex designs, identification of the appropriate level for tests and full reporting of outcomes |
| ☐ | ☒ | Estimates of effect sizes (e.g. Cohen's *d*, Pearson's *r*), indicating how they were calculated |

*Our web collection on statistics for biologists contains articles on many of the points above.*

## Software and code

Policy information about availability of computer code

| | |
|---|---|
| Data collection | Stimuli were rendered using the open-source modelling software Blender (v2.79). Stimulus presentation and data collection were controlled by a MATLAB script (release 2018b, Mathworks, Natick, MA) using the Psychophysics Toolbox (v3; Brainard, 1997). |
| Data analysis | Data analysis was performed using MATLAB scripts (release 2018b, Mathworks, Natick, MA). Code for analyses (including image analyses) are available on Zendo: DOI: 10.5281/zenodo.5080227. |

For manuscripts utilizing custom algorithms or software that are central to the research but not yet described in published literature, software must be made available to editors/reviewers. We strongly encourage code deposition in a community repository (e.g. GitHub). See the Nature Research guidelines for submitting code & software for further information.

## Data

Policy information about availability of data

All manuscripts must include a data availability statement. This statement should provide the following information, where applicable:
- Accession codes, unique identifiers, or web links for publicly available datasets
- A list of figures that have associated raw data
- A description of any restrictions on data availability

Psychophysics data and stimuli used in the experiments are available on Zendo: DOI: 10.5281/zenodo.5080227. The 3D meshes of the bunny and dragon objects were obtained from the Stanford 3D Scanning Repository and can be found under the following link: http://graphics.stanford.edu/data/3Dscanrep/.

October 2018

# Field-specific reporting

Please select the one below that is the best fit for your research. If you are not sure, read the appropriate sections before making your selection.

☐ Life sciences ☒ Behavioural & social sciences ☐ Ecological, evolutionary & environmental sciences

For a reference copy of the document with all sections, see nature.com/documents/nr-reporting-summary-flat.pdf

# Behavioural & social sciences study design

All studies must disclose on these points even when the disclosure is negative.

| | |
|---|---|
| Study description | The study had a within-subject experimental design, with quantitative data for all experiments. |
| Research sample | Participants were undergraduate students from the psychology programme at Justus Liebig University Giessen in Germany. Fifteen participants completed the free-naming experiment (Experiment 1, mean age: 23.7, female 80%, male 20% ). Eighty native-level German speakers participated in the 18-AFC experiment (Experiment 2, mean age 24.9, female: 83.3%, male: 16.7%), 22 participants took part in the gloss rating experiment (Experiment 3, mean age: 25.3, female: 58.3%, male: 41.7%), and 22 participants took part in the feature manipulation experiment (Experiment 4, mean age: 23.4 , female: 81.8%, male: 18.2%). Different participants were recruited for each experiment. |
| Sampling strategy | The experiment was advertised to students at Justus Liebig Univerity through the university's experimental volunteer system and participants were chosen on a first come first served basis. Sample size was chosen based on standards in the field, i.e., psychophysical studies of mid-level perception, and was slightly higher than this standard (e.g., 20 participants per experiment in Storrs, Anderson, & Fleming, 2021; Nat. Hum. Behav.). |
| Data collection | In the free naming task (Experiment 1), stimuli were projected onto a white wall in a classroom. Thick black cloth was used to block light from the windows, so that the only source of light came from the projected image. For all other experiments the stimuli were presented on a Sony OLED monitor running at a refresh rate of 120 Hz with a resolution of 1920 x 1080 pixels controlled by a Dell computer running Windows 10. Stimuli were viewed in a dark room at a viewing distance of approximately 60cm. The only source of light was the monitor that displayed the stimuli. Participants used mouse and keyboard presses to respond to stimuli presented on the screen. For Experiment 1 participants recorded their responses using pen and paper. Only experimenter and participant were present during the experiment.<br><br>In Experiment 1, two researchers were present in the same room as participants. In Experiments 2-4, a researcher was present in the same room as the participant. In all Experiments, researchers were not blinded to the study hypothesis, however our experiments did not involve assignment to groups (but within subject design). |
| Timing | Experiment 1: 3rd May 2018 (one day only); Experiment 2: 11th March - 14th May 2019; Experiment 3: 14th-20th August 2019; Experiment 4: 11th-21st November 2019 |
| Data exclusions | One participant was excluded from Experiment 3 because they did not understand the task instructions. Not understanding the task was a pre-established reason for excluding participants from analysis. |
| Non-participation | No participants dropped out/declinded participation. |
| Randomization | Participants were not allocated into experimental groups (within subjects design). |

# Reporting for specific materials, systems and methods

We require information from authors about some types of materials, experimental systems and methods used in many studies. Here, indicate whether each material, system or method listed is relevant to your study. If you are not sure if a list item applies to your research, read the appropriate section before selecting a response.

## Materials & experimental systems

| n/a | Involved in the study |
|---|---|
| ☒ | Antibodies |
| ☒ | Eukaryotic cell lines |
| ☒ | Palaeontology |
| ☒ | Animals and other organisms |
| ☐ | ☒ Human research participants |
| ☒ | Clinical data |

## Methods

| n/a | Involved in the study |
|---|---|
| ☒ | ChIP-seq |
| ☒ | Flow cytometry |
| ☒ | MRI-based neuroimaging |

# Human research participants

Policy information about studies involving human research participants

| | |
|---|---|
| Population characteristics | See above. |
| Recruitment | Participants volunteered via a university experimental volunteer system, and participants were chosen on a first come first served basis. Participants were chosen based on their availability at the time of data collection. Visual perception of materials likely varies somewhat with environmental and cultural exposure to different materials. We consider our sample of educated young adults to be representative of visual material perception within industrialised Western countries, but not necessarily representative of all humans.<br><br>Any potential self selection biases such as financial pressure, or motivation to gain experience in a psychological experience are not likely to impact the generalisability of the results, since our investigation focuses on basic perceptual mechanisms. |
| Ethics oversight | This study was approved by the local ethics review board of the Justus Liebig University Giessen (LEK FB 06) and strictly adhered to the ethical guidelines put forward by the declaration of Helsinki (2013). All participants gave written informed consent prior to the experiments and were told about the purpose of the experiments. All participants were compensated for their participation at a rate of 8 €/hour. |

Note that full information on the approval of the study protocol must also be provided in the manuscript.

