## [Peer Review File · Nature Human Behaviour]

Peer Review Information

Journal: Nature Human Behaviour

Manuscript Title: Material category of visual objects computed from specular image structure

Corresponding author name(s): Alexandra C. Schmid

Editorial Notes:

This manuscript has been previously reviewed at another journal that is not operating a transparent peer review scheme. This document only contains reviewer comments, rebuttal and decision letters for versions considered at Nature Human Behaviour.

Reviewer Comments & Decisions:

Decision Letter, initial version:

26th August 2021

Dear Dr Schmid,

Thank you once again for your manuscript, entitled "Material category of visual objects computed from specular image structure," and for your patience during the peer review process.

Your manuscript has now been evaluated by 2 reviewers, whose comments are included at the end of this letter. Although the reviewers find your work to be of interest, they also raise some concerns.

We are interested in the possibility of publishing your study in Nature Human Behaviour, but would like to consider your response to these concerns in the form of a revised manuscript that addresses all of the points raised, before we make a decision on publication.

In sum, we invite you to revise your manuscript taking into account all reviewer and editor comments. We are committed to providing a fair and constructive peer-review process. Do not hesitate to contact us if there are specific requests from the reviewers that you believe are technically impossible or unlikely to yield a meaningful outcome.

We hope to receive your revised manuscript within 12 weeks (3 months). We understand that the

COVID-19 pandemic is causing significant disruption for many of our authors and reviewers. If you cannot send your revised manuscript within this time, please let us know - we will be happy to extend the submission date to enable you to complete your work on the revision.

- Include a "Response to the editors and reviewers" document detailing, point-by-point, how you addressed each editor and referee comment. If no action was taken to address a point, you must provide a compelling argument. This response will be used by the editors to evaluate your revision and sent back to the reviewers along with the revised manuscript.
- Highlight all changes made to your manuscript or provide us with a version that tracks changes.

[REDACTED]

We look forward to seeing the revised manuscript and thank you for the opportunity to review your work. Please do not hesitate to contact me if you have any questions or would like to discuss these revisions further.

Sincerely,
Jamie

Dr Jamie Horder
Senior Editor
Nature Human Behaviour

REVIEWER COMMENTS:

Reviewer #1:
Remarks to the Author:

General comments

The authors showed in a number of analyses that object perception can be predicted from a subset of cues that are instilled in the images that human beings see. Although the manuscript lacks conceptual novelty, it represents a major evidence-based advance due to its rigour superseding the existing literature and significantly strengthen confidence in a scientific finding or convincingly falsify it. The authors showed that specular reflections are more important for the inference of material

characteristics of objects than was previously thought from the very limited amount the previous literature by other authors in their research field. This seems to be the main key point of the whole study. The authors do go on to mention the important part that color plays in this problem of classification.

The methods seems to be soundly carried out with great rigor.

The results are deeply analysed but I wonder about the presentation of some results with the examples given.

Also, do the authors have calculations of inter-rater reliability that they themselves can disclose? There is mention of glossiness but I am wondering about consistency in judgements of material classification.

I had some questions about possible bias in the presentation of the results to the readers (please refer to the more specific comments listed further down).

There has been a lot of progress in deep learning for categorical sorting of images into materials. I wonder whether the authors can include some discussion about this important field outside their own.

Specific comments

Overall, the literature review seems comprehensive.

What is intended by "including many that should be defined by more complex scattering functions." This may refer to the brdf of shimmer paints?

I do not understand this sentence "A likely reason for this discrepancy is that studying properties like colour and gloss seems more tractable than discovering the necessary and sufficient conditions for recognising the many material classes in our environment." what is discrepant. Readers may not follow?

I do not understand what shape is a mid-level property. This is in my understanding a real world property.

"A feedforward view of neural processing (i) assumes that categories are defined by combinations of estimated mid-level properties like gloss, colour, and shape, etc.,"

The swatches in fig.1 are colored. I do not see how the rest of the figure relates to the color of the objects that appear to be important for their theory.

Is the following the same as the scattering the authors refer to in the abstract "However, reflections from plastic can look tinted if the prevailing illumination is coloured." I would believe this tinting emerges from blending light with the plastic material.

$n = 15 \gg$ a power analysis should be provided.

Fig.3 I believe A[right] is velvet looking. The middle image looks to change in specular reflectance and lighter shading. Comparing the left and right images is interesting. The purpose of the middle image is not clear.

I very much like the factor analysis data. This is very important. It shows the authors claim clearly. Is it possible the subjects choose the material class that is closest to what see but this is still rather different to what they actually feel?

The figures allow material examples to be compared by the reader but the examples are not similar (for example dragon and bunny). Is there some bias towards the reader here. I wonder what melted chocolate looks like when contrasted to fresh non-melted chocolate. This is a good example where the same specular and shading parameters could make the subject see different materials depending on the color of the object. I note that the authors say further along "For example, "chocolate" and "plastic" could have similar contrast and sharpness of specular highlights (similar "gloss types") and the different labels might result from a cognitive decision by participants based on body colour (brown versus yellow). However, we cannot think of a principled reason why a cue's influence on material category would affect how people choose to use that cue for gloss judgments."

Comparison would be easier when the objects are the same. This is how computer vision readers read studies about material classification.

There is also a lack of equations used to describe calculation concepts.

"the stimuli nevertheless convincingly resemble silk-like, porcelain-like, wax-like, brushed metal-like, etc."

Again, I wonder if these are the best choices for the reader and subject in the study. What type of assurance do the authors have to suggest no such bias exists in their study.

Is there any evidence of semantic input to the fact that "materials were differentiated directly based on image-measurable specular reflection cues, suggesting that material qualities like gloss are a perceptual outcome with – rather than a component dimension of – our holistic impressions"?

I think the argument could be made more strongly for the reader to understand. And, I wonder about whether the results would be different with a free choice experimental design instead of a forced choice experimental design. These two both would use qualitative methods informed by the classing of object properties that the authors are studying.

"We found that inter-subject agreement for gloss ratings was high (median $r = 0.70$)....We investigated whether the wide gloss distributions could be caused by the continuous nature of the material dimensions (e.g., perhaps more metallic-looking materials are glossier, more rubber-looking materials are less glossy, etc.)....half showing no significant correlation between gloss level and score (black coefficients), suggesting that perceived gloss level is overall not a good indicator of how well stimuli load onto a material dimension."

I wonder whether this sort of inter-rater agreement analysis would transfer to material classes that the authors suggest further along in their manuscript.

Reviewer #2:

Remarks to the Author:

Review of "Material category of visual objects computed from specular image structure"

Reviewer: Michael Landy

This paper uses free naming of materials and later a categorization task using computer rendered stimuli to ask whether material perception follows intermediate-level vision in a feed-forward manner (image -> perception of gloss, etc. -> material categorization) and finds this not to be the case. Although the evidence is a bit indirect, the story is pretty convincing and interesting. Using Blender-generated stimuli and a fairly ad hoc set of rendering parameters, they manage to pull a lot out of this fairly qualitative story. In short, I think it's a strong paper, heavy on interesting approaches to data analysis to wring a story out of fairly complex data. I think most of my comments are about clarifying the manuscript.

Comments, big and small (Line #):

289: "image-computable measures": As I understood it, much of these measures are computed using the separated diffuse and specular images, which aren't available for real-world images. (Although, my student Yunxian Ho made some lovely images of nectarines once, using crossed polarizers to pull out the diffuse reflection, which I got to use for my commentary on Motoyoshi et al., 2007 ;^)

332: "as would be expected if the visual system computed surface gloss from image cues independent of material class": Well, as would be expected if gloss was computed LINEARLY from these cues. This particular analysis does not disprove a nonlinear integration of cues.

Figure 5E: Are these R^2 values from all the data or are they cross-validated? There's nothing all that surprising about increasing R^2 values when more regressors are added.

378: I'm mainly familiar with the two-class version of LDA. With your 18 classes, this could be used to discriminate each single category from the rest (pooled). But, you claim chance rates of 1/18, so clearly you are using some form of LDA to do the 18-way classification. I don't know how standard it is to use LDA in that way (I assume it isn't the same as nearest neighbor, after whitening a common covariance, which is an equal-prior Bayesian approach), and so it would be good to cite or describe what you did.

413-414: If this combination can be used to discriminate, then it should be visible in the weights of your LDA solution, yes?

670-671: I have no idea what is meant by the "tangent direction" (tangency of what relative to what?) nor what it means to set it radial. Please clarify for the non-cognoscenti.

725: Is this calculation a dot product with the input image to replace the center pixel? Are the weights normalized in any way to, e.g., make the sum of the weights from an input pixel be one? Clarify, please.

752: "Smooth Hermite interpolation:": A Hermite interpolation goes through a set of values with pre-specified derivatives at each. You don't specify what points and derivatives each $f()$ is forced to match.

759: " n_m is a manually-set direction...": This seems pretty ad hoc. Is there a physical model that

corresponds to this manipulation? Similarly (776-778), I had no idea why these 4 directions for n_m were chosen for the four combinations of object and light field.

908-909: "This was guided by correlations...": It took me a while to figure out what pairs of vectors were correlated here. I think a revision to the text that clearly states what the pairs are that are correlated would be useful (what the elements of the vector are indexed by, and what the values are). You say some of this, but in a way that's hard to parse.

971-973: Please clarify what is meant by this matching of total reflectance; I couldn't figure it out.

989: "average colour saturation": Throughout, your computations are in RGB, not in a physical model (as a function of wavelength), as is generally done in computer graphics. Some clear mention of that would be nice, and here, your definition of "saturation" as a function of RGB should be provided (yes, there are standard versions of this function, but they are all a bit ad hoc).

1000: "stimuli that were allocated a dimension". I'm not particularly familiar with factor analysis, but my understanding is that the identified factors are only known up to an orthogonal transformation, so that you must have rotated the solution to make it come out so that factors sort of corresponded to stimuli, right? So "allocated a dimension" is putting the onus on the data for yielding single-stimulus factors, when this is the result of an arbitrary rotation, right? Also, PCA yields unique answers such that extracting 8 PCs instead of 7, just adds one new PC (the others stay the same). Is it also the case that factor analysis for $n+1$ factors adds one new dimension, but the n -dimensional subspace is still in the $n+1$ factor solution (i.e., factor analysis doesn't really depend on the number of retained factors. Relates to Figure S7)? Your paper reads like the arbitrary choice of the number of factors is expected to behave differently for different values of n , which isn't obvious to me. Again, I admit that I've never really studied factor analysis.

Legend, Figure S1: "Only reflectance parameters were manipulated". Are there any parameters other than reflectance parameters?

Legend, Figure S4: Again, it's unobvious from this legend as to what pair of vectors are correlated to produce each entry in this matrix.

Figure S8: (1) You ended up with "Gold metals" due to the arbitrary choice of a yellow color for the colored stimuli. Clearly, the outcome would have been different for other colors. You barely discuss the fact that the set of materials was highly dependent on this arbitrary choice. This is not a problem for the study, but should have been overt early on. (2) Are the backgrounds straight from the light fields or are they something else. Roland Fleming pasted rendered objects onto a background that wasn't lit by the light field, which always seemed problematic to me. So what are these backgrounds? (3) What exactly does it mean for a stimulus to be from an emergent dimension? Please clarify.

Figure S14: I didn't understand this figure nor its legend at all. Really couldn't parse it and figure out what you did nor what it meant.

RESPONSE TO REVIEWERS

We thank the reviewers for taking the time to read and comment on the manuscript. We provide a point-by-point response (black) to each comment (blue) below.

REVIEWER COMMENTS:

Reviewer #1:

General comments

The authors showed in a number of analyses that object perception can be predicted from a subset of cues that are instilled in the images that human beings see. Although the manuscript lacks conceptual novelty, it represents a major evidence-based advance due to its rigour superseding the existing literature and significantly strengthen confidence in a scientific finding or convincingly falsify it. The authors showed that specular reflections are more important for the inference of material characteristics of objects than was previously thought from the very limited amount the previous literature by other authors in their research field. This seems to be the main key point of the whole study. The authors do go on to mention the important part that color plays in this problem of classification.

We thank the reviewer for the time and effort reading and evaluating our work and for their general positive assessment. We would, however, like to take the opportunity to respond to the comment about conceptual novelty. The reviewer describes the theoretical implication of our work as being 'that object perception can be predicted from a subset of cues that are instilled in the images that human beings see'. We believe this is a little too broadly construed.

Instead, in our opinion, the novel insights are that changes in cues that supposedly only affect the glossy appearance of an object *also* determine the material category and there is an interesting *reciprocal relationship* between the perceived material category and the interpretation of cues (to gloss).

In that sense not only are specular reflections more important for the inference of material characteristics of objects than was previously thought, but specifically we show that the salient image structure (e.g., from specular reflections or the interaction of reflections with other sources of image structure) may directly contain diagnostic information about material *class*, which is a novel insight.

The methods seem to be soundly carried out with great rigor.

Thank you.

The results are deeply analysed but I wonder about the presentation of some results with the examples given.

We respond to this in the specific comments below.

Also, do the authors have calculations of inter-rater reliability that they themselves can disclose? There is mention of glossiness but I am wondering about consistency in judgements of material classification.

Consistency between participants of material categorization has been directly factored into the computation of category profiles in Experiment 2. In Figure 3B, the number in each field is the sum of the confidence rating for a particular category response. 'Empty' means, no single participant thought this category label is suitable for the presented stimulus. Alternatively, if all observers (20 per stimulus) chose the same category label with maximum confidence (3) this value would be 60, if all chose this category with medium confidence (2), this value would be 40 and if the category was chosen by all observers with the lowest confidence it would be 20. While we cannot tell from the plot the exact mixture of high, medium, and low confidences that went into the values of the category profiles in Figure 2B, the values do give a rough general estimate of interobserver consistencies: the higher the value the larger the consistency.

I had some questions about possible bias in the presentation of the results to the readers (please refer to the more specific comments listed further down).

We respond below.

There has been a lot of progress in deep learning for categorical sorting of images into materials. I wonder whether the authors can include some discussion about this important field outside their own.

Thank you for this suggestion and we agree that the manuscript would benefit from a discussion of DNNS. We have included the following on page 22:

"Recently, unsupervised deep neural networks (DNNS) have generated excitement as a potential tool to uncover purported mid-level visual processes, for example by providing evidence that the brain could potentially spontaneously discover the existence of properties like gloss, illumination direction, and shape without prior knowledge about the world, but by learning to efficiently represent variations in image structure produced by these properties (Storrs et al., 2021).

However, here we showed that with more complex stimuli image information cannot be neatly mapped onto perceptual properties like gloss; this mapping is constrained by the qualitative, or categorical appearance of surfaces. Some material classes occupy relatively homogeneous feature spaces (e.g., gold) compared to other, more diverse classes (e.g., plastic), which likely emerges through behavioural relevance. Thus, for unsupervised learning to generate useful hypotheses about perceptual processes, future work needs to consider how the behavioural relevance of stimuli might influence spontaneous learning about statistical image variations."

Specific comments

Overall, the literature review seems comprehensive.

Thank you.

What is intended by "including many that should be defined by more complex scattering functions." This may refer to the brdf of shimmer paints?

We specifically meant to refer to subsurface scattering, which is usually thought to be required to depict the appearance of e.g. wax or soap materials. We have now clarified this in the text on page 2.

I do not understand this sentence "A likely reason for this discrepancy is that studying properties like colour and gloss seems more tractable than discovering the necessary and sufficient conditions for recognising the many material classes in our environment." what is discrepant. Readers may not follow?

We apologize that this was confusing. What we meant was the disparate focus of the literature on material properties versus classes. We have now clarified this in the text on page 3.

I do not understand what shape is a mid-level property. This is in my understanding a real world property. "A feedforward view of neural processing (i) assumes that categories are defined by combinations of estimated mid-level properties like gloss, colour, and shape, etc.,"

It is true that shape is both a property of the real world and also a perceptual property. We have now clarified that we mean "apparent shape" here (page 5, Figure 1 caption).

The swatches in fig.1 are colored. I do not see how the rest of the figure relates to the color of the objects that appear to be important for their theory.

Figure 1 consists of 4 panels A.-D. panel A is an illustration how materials' specular reflectance can produce characteristic image structure, panel B. contrasts processing pipelines according to feedforward and simultaneous process theories, panel C illustrates how a manipulation of specular reflectance parameters (as from a rendering model, e.g. Ward) not only leads to apparent changes in gloss but also to changes in material category, and panel D lists the reflectance parameters that were manipulated in the experiments and some examples of how their manipulation affects appearance.

We are not sure which panel the reviewer is referring to, but we are interpreting this comment as that the reviewer thinks that the role of color changes in material classification is not illustrated, whereas how changes in specular parameters might affect material classification is illustrated in Panels C and D.

While it is true that panel C focuses on the effects of changes in specular parameters, Panel D actually does show an example of how color changes affect material category, say from a more porcelain-like appearance on the top left to a more melting-chocolate-like appearance on the top right. To emphasize the role of color in our model we add the following sentence to the caption, under the section of panel C:

'note that for some materials like gold, colour information also contributes to perceptual classification'

Is the following the same as the scattering the authors refer to in the abstract "However, reflections from plastic can look tinted if the prevailing illumination is coloured." I would believe this tinting emerges from blending light with the plastic material.

The coloring of reflections here described corresponds to a tinting due to the color of the lighting, since plastic materials do not produce colored reflections. It is thus not the same as the scattering functions described in the abstract, which rather point to materials that exhibit translucent effects (wax, porcelain, etc). The end of the caption of Figure 2 has been rewritten to clarify its meaning:

"The brightness and colour saturation inside and outside of the highlight region is also determined by interactions between a surface's absorption/reflectance properties, 3D shape, and the illumination field. For example, specular reflections from dielectric materials like plastic preserve the spectral content of incident light; hence they look tinted only if the prevailing illumination is itself coloured. In contrast, coloured metals affect the tint of reflected light."

n = 15 >> a power analysis should be provided.

We think the reviewer is referring to Experiment 1 where 15 participants were asked to freely generate names of materials each object was made from. The purpose of this experiment was to generate terms in an unbiased manner, and no conditions were compared / no inferential statistics were performed. We don't fail to detect an effect, which is what a power analysis would be useful for. Furthermore, one usually refrains from performing posthoc power analyses, and we are unsure what we would gain from (or how to even go about) one in this particular case.

Fig.3 I believe A[right] is velvet looking. The middle image looks to change in specular reflectance and lighter shading. Comparing the left and right images is interesting. The purpose of the middle image is not clear.

The purpose of this figure is to make the point that multiple semantic terms can describe the same qualitative visual appearance of a material. For example, (A) could be labeled as either melted chocolate or mud based on a cognitive decision by participants (rather than chocolate and mud being visually qualitatively different).

Furthermore, the fact that participants categorized the middle image as wax/soap is what makes it interesting, since such materials usually require subsurface scattering, which is entirely absent here. Instead, categorical appearance is mainly due to specular structure.

Having said that, we do agree that it would be more compelling to show examples of different categories where specular image structure changes without a change in base colour. We have now replaced the wax/soap example with an exemplar from the category latex/rubber, so that only changes in the appearance of specular reflections are depicted.

I very much like the factor analysis data. This is very important. It shows the authors claim clearly.

Thank you!

Is it possible the subjects choose the material class that is closest to what they see but this is still rather different to what they actually feel?

This is a very important point, and this was one specific reason why we used confidence ratings. We used these to weigh a given category choice.

The figures allow material examples to be compared by the reader but the examples are not similar (for example dragon and bunny). Is there some bias towards the reader here.

We believe the reviewer is referring to Figure 4 which shows typical stimulus for each material category across both shapes and light fields. We agree with the reviewer that it might make more sense to show exemplars from the same shape and light field, so that the reader can focus on changes in reflectance properties across categories. We have changed the examples depicted Figure 4B accordingly.

Supplementary Figure 8 shows examples from each shape x light field combination, as is pointed out in the caption of Figure 4.

I wonder what melted chocolate looks like when contrasted to fresh non-melted chocolate. This is a good example where the same specular and shading parameters could make the subject see different materials

depending on the color of the object. I note that the authors say further along “ For example, “chocolate” and “plastic” could have similar contrast and sharpness of specular highlights (similar “gloss types”) and the different labels might result from a cognitive decision by participants based on body colour (brown versus yellow). However, we cannot think of a principled reason why a cue’s influence on material category would affect how people choose to use that cue for gloss judgments.”

Now that we have changed exemplars in Figure 4 to the same shape and light field (see above comment), such comparisons can be made directly.

Comparison would be easier when the objects are the same. This is how computer vision readers read studies about material classification.

We have now done this (see our comments above).

There is also a lack of equations used to describe calculation concepts.

We are not sure which equations the reviewer is missing. We describe the computation of gloss features on page 33 and in Supplementary Figure 9, as well as image-based cue manipulation in detail on pages 25-28.

“ the stimuli nevertheless convincingly resemble silk-like, porcelain-like, wax-like, brushed metal-like, etc.”

Again, I wonder if these are the best choices for the reader and subject in the study. What type of assurance do the authors have to suggest no such bias exists in their study.

We appreciate the reviewer’s concern. We would like to point out that we, the authors, did not just come up with these categories but that instead they were derived experimentally from the responses in a free naming task (Experiment 1). That should have safeguarded us against some potential selection biases.

Please also see page 7: ‘We collected unbiased participant-generated category terms for each stimulus using a free-naming task (Experiment 1) where participants (n=15) judged what material each object was made from with no restrictions.’ And we continue describing in detail how the 18 categories that were used in Experiment 2, were derived. Please also see Supplementary Figure 2-4 for further details of this analysis.

We also would like to re-state that we believe that labels are only relevant to the extent that they capture qualitative perceptual differences in visual material appearance.

Is there any evidence of semantic input to the fact that “materials were differentiated directly based on image-measurable specular reflection cues, suggesting that material qualities like gloss are a perceptual outcome with – rather than a component dimension of – our holistic impressions”?

I think the argument could be made more strongly for the reader to understand. (the paragraph about cognition)

The reviewer refers to the sentence on p22: ‘Specifically, we found that gloss was not a good indicator of a material’s class; instead, materials were differentiated directly based on image- measurable specular reflection cues, suggesting that material qualities like gloss are a perceptual outcome with – rather than a component dimension of – our holistic impressions.’

This statement is indirectly implying a prediction of the feedforward hypothesis: 'we should be able to identify image cues that predict gloss perception, and the material categories from Experiment 2 should be associated with a particular level of gloss.' (p.12)

However, this is not what we found. As stated in the text on p. 11: 'the material dimensions defined in Experiment 2 were not associated with a particular level of gloss, contrary to what would be predicted by the *feedforward hypothesis*. Instead, stimuli from the same material class exhibited a wide distribution of gloss levels, and stimuli from very visually distinct classes like ceramic and (gold and uncoloured) metals had completely overlapping gloss distributions (histograms in Figure 5B).' This is also reflected in the Representational Similarity results in Figure 7 the prediction of material category profile by gloss ratings+color features is inferior ($R^2=.23$) to the prediction of category profile directly from image cues ($R^2=.34$). If the feedforward model was true, we would expect a relatively stronger association between gloss rating and material category profile, because gloss would be a somewhat reliable component of material class.

And, I wonder about whether the results would be different with a free choice experimental design instead of a forced choice experimental design. These two both would use qualitative methods informed by the classing of object properties that the authors are studying.

The reviewer is concerned that the forced-choice design of Experiment 2 might have affected these results. However, based on other comments, we think this reviewer may have missed the fact that 18 categories participants chose from in Experiment 2 were derived empirically with a free-choice experiment (Experiment 1). Therefore, we are confident that we captured the categorical space covered by our stimuli and believe that adding more or different categories is unlikely to substantially change the results, especially since the factor analysis ended up reducing the number of dimensions needed to account for the results. Furthermore, the confidence of category fit, i.e., how well the participant thought the category captures the stimulus, was factored into the analyses). Please also see Figure 3B.

Lastly, we also would like to repeat our statement from above that we believe that the specific labels are only relevant to the extent that they capture qualitative perceptual differences in visual material appearance.

"We found that inter-subject agreement for gloss ratings was high (median $r = 0.70$) We investigated whether the wide gloss distributions could be caused by the continuous nature of the material dimensions (e.g., perhaps more metallic-looking materials are glossier, more rubber-looking materials are less glossy, etc.) half showing no significant correlation between gloss level and score (black coefficients), suggesting that perceived gloss level is overall not a good indicator of how well stimuli load onto a material dimension."

I wonder whether this sort of inter-rater agreement analysis would transfer to material classes that the authors suggest further along in their manuscript.

We have responded to this suggestion for interobserver correlation for categorical responses above.

Reviewer #2:

Remarks to the Author:

Review of "Material category of visual objects computed from specular image structure" Reviewer:

Michael Landy

This paper uses free naming of materials and later a categorization task using computer rendered stimuli to ask whether material perception follows intermediate-level vision in a feed-forward manner (image -> perception of gloss, etc. -> material categorization) and finds this not to be the case. Although the evidence is a bit indirect, the story is pretty convincing and interesting. Using Blender-generated stimuli and a fairly ad hoc set of rendering parameters, they manage to pull a lot out of this fairly qualitative story. In short, I think it's a strong paper, heavy on interesting approaches to data analysis to wring a story out of fairly complex data. I think most of my comments are about clarifying the manuscript.

Thank you for this overall positive evaluation.

Comments, big and small (Line #):

289: "image-computable measures": As I understood it, much of these measures are computed using the separated diffuse and specular images, which aren't available for real-world images. (Although, my student Yunxian Ho made some lovely images of nectarines once, using crossed polarizers to pull out the diffuse reflection, which I got to use for my commentary on Motoyoshi et al., 2007 ;^)

It is true that the measures we provide in the paper are not directly image-computable, since we relied on specific render outputs. A separation into diffuse and specular contributions simplifies our computational analysis in practice, since there is no established method to separate such contributions from a single image as far as we know. More importantly, our point was not to suggest that the human visual system (HVS) necessarily has to separate an image into diffuse and specular components, but rather that the HVS may attend to specific diffuse or specular image features, in cases where they can be distinguished. To avoid confusion, we removed the term "image-computable measures" and changed the text to: "here we operationalised the cues using objective, image-based measures computed from dedicated render outputs".

332: "as would be expected if the visual system computed surface gloss from image cues independent of material class": Well, as would be expected if gloss was computed LINEARLY from these cues. This particular analysis does not disprove a nonlinear integration of cues.

The reviewer is correct, and we have now added the word "linearly" to the sentence on page 12

Figure 5E: Are these R^2 values from all the data or are they cross-validated? There's nothing all that surprising about increasing R^2 values when more regressors are added.

The reviewer is correct and we have now changed the R^2 values to adjusted R^2 , which accounts for different number of predictors in each model (the results remain unchanged).

378: I'm mainly familiar with the two-class version of LDA. With your 18 classes, this could be used to discriminate each single category from the rest (pooled). But, you claim chance rates of $1/18$, so clearly you are using some form of LDA to do the 18-way classification. I don't know how standard it is to use LDA in that way (I assume it isn't the same as nearest neighbor, after whitening a common covariance, which is an equal-prior Bayesian approach), and so it would be good to cite or describe what you did.

The reviewer is asking about the linear discriminant analysis (LDA) that classified materials based on linear combinations of image cues. We have added more details about this analysis on page 33:

'Materials were classified based on linear combinations of image cues using multi-class LDA in Matlab (<https://www.mathworks.com/help/stats/discriminant-analysis.html>), which finds a set of linear combinations that maximizes the ratio of between-class scattering to the within-class scattering.

The matlab function `fitdiscr` was used to fit a discriminant analysis classifier to the training data (regularised linear discriminant analysis where all classes have the same covariance matrix). To train a classifier, the fitting function estimates the parameters of a Gaussian distribution for each class (<https://www.mathworks.com/help/stats/creating-discriminant-analysis-model.html>).

Test data labels were predicted with the Matlab function `predict`, where the trained classifier finds the class with the smallest misclassification cost (<https://www.mathworks.com/help/stats/prediction-using-discriminant-analysis-models.html>).

413-414: If this combination can be used to discriminate, then it should be visible in the weights of your LDA solution, yes?

Yes. We now plot the weights for each image cue on each Linear Discriminant in the new Supplementary Table 1.

670-671: I have no idea what is meant by the "tangent direction" (tangency of what relative to what?) nor what it means to set it radial. Please clarify for the non-cognoscenti.

We apologize that this wasn't clear and have now provided further details on the tangent direction in the methods section on page 24:

"Higher values give elongated highlights along a surface tangent direction given by a tangent field. This field is obtained per object using the spherical mapping functionality of Blender. Internally, it works in two steps. First, a radial pattern of directions is defined on a 3D sphere enclosing the object, by assigning the direction of the meridian going through each spherical point. Such a pattern has singularities at sphere poles, where all directions converge. Second, each point p on the object is mapped to a point q on the enclosing sphere by tracing a ray from the object center through p . The direction at q is then projected on the local tangent plane at p and renormalized to yield the local tangent. Note that singularities remain singularities after projection."

725: Is this calculation a dot product with the input image to replace the center pixel? Are the weights normalized in any way to, e.g., make the sum of the weights from an input pixel be one? Clarify, please.

We have added details on the cross bilateral filtering process in the methods section on page 26:

"The cross bilateral filter consists in an image convolution with a modified blur kernel: it not only weights the contribution of neighbor pixel colors given their distance (the spatial weight), but also based on additional pixel-wise information (the range weight). The two weights are combined multiplicatively, and the blur kernel is systematically normalized so that the sum of combined weights is one."

752: "Smooth Hermite interpolation": A Hermite interpolation goes through a set of values with pre-specified derivatives at each. You don't specify what points and derivatives each $f()$ is forced to match.

Thank you for that remark. We used this interpolation for producing the specific effects of velvet/satin. After normal-based blurring, the specular component is multiplied by a first mask m_1 that is computed via a

non-linear mapping of the luminance I of the specular component. Here, a specific type of Hermite interpolation is used, which is called the "smooth step" function in Computer Graphics (<https://en.wikipedia.org/wiki/Smoothstep>). We have now added a reference to this function to the text on page 26. Supplementary Figures 17 and 16 show mapping function and mapping results, respectively.

759: "n_m is a manually-set direction...": This seems pretty ad hoc. Is there a physical model that corresponds to this manipulation? Similarly (776-778), I had no idea why these 4 directions for n_m were chosen for the four combinations of object and light field.

The main idea is to position n_m in the direction of the main highlight. This could be done automatically, but since that direction is fixed per scene, we have found it sufficient to indicate it manually. It does not correspond to a physical model, but instead mimics the unconventional darkening observed in images of velvet materials at locations where the main highlight should occur.

We have added this information to the manuscript on page 26.

908-909: "This was guided by correlations...": It took me a while to figure out what pairs of vectors were correlated here. I think a revision to the text that clearly states what the pairs are that are correlated would be useful (what the elements of the vector are indexed by, and what the values are). You say some of this, but in a way that's hard to parse.

We rewrote this passage, which hopefully improves readability:

'We decided to retain the category terms from the free-naming task that at least 5 out of 15 participants used. This was an arbitrary cut off with the aim of being quite inclusive while at the same time maintaining some consensus among participants. Supplementary Figure 2 shows the 29 category terms that survived this cutoff. Subsequently, two of the authors (AS and KD) worked together to reduce this set of category terms. For example, materials that were visually or semantically similar were combined (specifically: fabrics, including velvet and silk; unglazed porcelain, stone, and chalk; latex and rubber; wax and soap; brass, bronze, and copper; iron and chrome; and silver and aluminium). To have some numerical guidance for this reduction we computed the correlation between each pair of categories, calculated from the number of participants that used each term for each stimulus (Supplementary Figure 4). That is, for each pair of categories, correlated vectors were indexed by stimulus number with the values being number of observer responses.

971-973: Please clarify what is meant by this matching of total reflectance; I couldn't figure it out.

We have rewritten some of this paragraph in an attempt to increase clarity (page 32):

"*Extra rendered images.* For each stimulus, a purely specular image was rendered, which had the same specular reflectance as the original (full rendered) image but with the diffuse component turned off. Two purely diffuse images were rendered, which we call "diffuse image 1" (used to calculate *coverage*) and "diffuse image 2" (used to calculate *sharpness* and *contrast*; first column in Supplementary Figure 9). Kim et al. (2012) separated specular highlights and lowlights by subtracting an image of a rendered glossy surface (with a specular and diffuse component) from an "equivalent" fully diffuse image with the same total reflectance as the glossy surface (i.e., the surfaces reflected the same amount of light in total; for the diffuse surface light was scattered equally in all directions, and for the glossy surface some light was reflected specularly with the rest scattered diffusely). Kim et al. used the Ward model (Ward, 1994) where the total reflectance could be easily matched between glossy and diffuse renderings because specular

reflectivity is constant at all viewing angles. Since the Principled BSDF simulates the Fresnel effect (whereby specular reflectance increases at grazing viewing angles, depending on the index of refraction), the diffuse (1) renderings were matched to the facing (along normal) reflectivity of the purely specular renderings. The second diffuse image (diffuse image 2) was created by rendering only the diffuse component of the original (full rendered) stimulus, with the specular component turned off.”

989: "average colour saturation": Throughout, your computations are in RGB, not in a physical model (as a function of wavelength), as is generally done in computer graphics. Some clear mention of that would be nice, and here, your definition of "saturation" as a function of RGB should be provided (yes, there are standard versions of this function, but they are all a bit ad hoc).

We now mention our computations are in RBG on page 33, and state how we calculate saturation:

“Cues. Image cues were calculated using RGB images. Coverage was defined as the proportion of object pixels that were calculated to be specular highlights (excluding lowlights). Contrast was the sum of root-mean-squared (RMS) contrast of extracted specular reflections at different spatial frequency bandpasses. Sharpness of extracted specular reflections was calculated for each pixel within the highlight regions using a measure of local phase coherence (Hassen, Wang, & Salama, 2013), then these values were averaged. Highlight saturation and lowlight saturation were calculated as the average colour saturation of pixels within the highlight region, and outside of the highlight region (which we call the lowlight region), respectively, as measured by the `rgb2hsv` function in MATLAB (release 2018b, Mathworks, Natick, MA), which converts the red, green, and blue values of an RGB image to hue, saturation, and value (HSV) values of an HSV image (<https://www.mathworks.com/help/matlab/ref/rgb2hsv.html>). Lowlight value was calculated as the average value of pixels in the lowlight region (also using the `rgb2hsv` MATLAB function). A root transformation on sharpness and contrast was applied to linearize the relationship of these cues to gloss ratings.”

1000: "stimuli that were allocated a dimension". I'm not particularly familiar with factor analysis, but my understanding is that the identified factors are only known up to an orthogonal transformation, so that you must have rotated the solution to make it come out so that factors sort of corresponded to stimuli, right? So "allocated a dimension" is putting the onus on the data for yielding single-stimulus factors, when this is the result of an arbitrary rotation, right? Also, PCA yields unique answers such that extracting 8 PCs instead of 7, just adds one new PC (the others stay the same). Is it also the case that factor analysis for $n+1$ factors adds one new dimension, but the n -dimensional subspace is still in the $n+1$ factor solution (i.e., factor analysis doesn't really depend on the number of retained factors. Relates to Figure S7)? Your paper reads like the arbitrary choice of the number of factors is expected to behave differently for different values of n , which isn't obvious to me. Again, I admit that I've never really studied factor analysis.

One aspect of this question is the reviewer trying to understand how PCA and factor analysis differ. All the things that they think are true of PCA are also true of factor analysis, the only difference is that the data being analyzed is the common variance between the variables (covariances) rather than the total variance of each variable.

Regarding the question about the solution changing when we retain n or $n+1$ factors: the answer is that the addition of another dimension/factor should not change the solution. However, there is a difference

between the solution/factor analysis changing depending on the number of factors we retain versus the category that each stimulus is allocated changing depending on the solution.

Yes, the reviewer is correct that the factor analysis is putting the onus on the data to yield a single-stimulus factor (i.e., we end up using the data to uniquely categorize each of our stimuli into a single category). This is the point. We categorized our stimuli by allocating them to the highest loading factor, and ignored how high the loadings were to any of the other factors.

e.g.

Example 1

Factor	I	II	III
Stimulus1	1	0	0
Stimulus2	0	1	0
Stimulus3	0	0	1
Stimulus4	0	0	1

vs.

Example 2

Factor	I	II	III
Stimulus1	.8	.6	0
Stimulus2	0	.8	0
Stimulus3	0	.6	0
Stimulus4	.6	0	.8

In Example 1 but not Example 2:

- each stimulus has high loadings on one factor only.
- each factor has high loadings for only some of the stimuli.

Example 2 seems to be the ambiguous situation that the reviewer might be worried about, and we acknowledge that this is reasonable worry/question, and a potential limitation of analyzing the results in this way. This is one reason why we included additional analyses like the RSA to compare differences in material appearance within a continuous rather than categorical space (Figure 7).

Legend, Figure S1: "Only reflectance parameters were manipulated". Are there any parameters other than reflectance parameters?

Yes. There are two additional ones that are strictly speaking not reflectance parameters: subsurface scattering, transmittance.

Legend, Figure S4: Again, it's unobvious from this legend as to what pair of vectors are correlated to produce each entry in this matrix.

Yes. This was not clear and we have added more information in the caption (see response above).

Figure S8: (1) You ended up with "Gold metals" due to the arbitrary choice of a yellow color for the colored stimuli. Clearly, the outcome would have been different for other colors. You barely discuss the fact that the set of materials was highly dependent on this arbitrary choice. This is not a problem for the study, but should have been overt early on.

In addition to bringing this point up in the General Discussion and also when we first talk about the image cues that define the categories (page 12), we have now also added text in two additional places:

Figure 1: “note that for some materials like gold, colour information also contributes to perceptual classification.”

Main text page 8: “Note that that some categories (like gold metals) occurred due to our arbitrary choice of yellow for the coloured stimuli, and the outcome would be different for other colours.

Nevertheless, as we show later, such materials are additionally defined by specular image structure, and colour information alone is not sufficient to discriminate materials.”

(2) Are the backgrounds straight from the light fields or are they something else. Roland Fleming pasted rendered objects onto a background that wasn't lit by the light field, which always seemed problematic to me. So what are these backgrounds?

The backgrounds are straight from the light fields. We have added a note about this to the methods section on page 25.

(3) What exactly does it mean for a stimulus to be from an emergent dimension? Please clarify.

Emergent means that these categories were obtained from a factor analysis on the stimulus profiles shown in Figure 3B. So in a sense, they are derived from visual feature measurements.

Figure S14: I didn't understand this figure nor its legend at all. Really couldn't parse it and figure out what you did nor what it meant.

We agree that this figure is confusing, and have now simplified it. This figure is basically about the numerical ‘success’ of our image manipulations. We have completely rewritten the figure caption, got rid of Panel B, D, and E, Panel C, and added Supplementary Tables 2 and 3 and hope this clarifies things.

‘Supplementary Figure 14. Visual feature manipulations. In order to test whether the visual gloss features we identified were indeed diagnostic for material category, in Experiment 4 we generated a new stimulus set through image feature manipulation (also see Figure 8 in the manuscript). (A) shows results for the “simple” manipulation condition, in which all *Gamma* (i.e., non-linear) parameters were set to 1 and the special velvet filter was discarded. (B) shows the manipulation results of the “complex” manipulation condition, which involved additional manipulations of the *sGamma*, *dGamma*, *sBoost* and *dBoost* parameters. The starting point for this new set was a rendered base image, which had been categorized as ‘glazed ceramic’, with a grayscale *Base Color* of 0.1, a *Specular* level of 0.3, a *Specular Tint* of 0, a *Roughness* of 0, and an *Anisotropic* level of 0 (see Figure 8A). By using an approach similar to compositing techniques used in visual effects (Birn, 2014) we generated all remaining material categories by manipulating our visual features (see Supplementary Tables 2 and 3), following the steps outlined in the methods section. (A.) Values on the x-axes correspond to the emergent material category obtained from factor analysis on the stimulus profiles (Figure 3B). The categories are in same order as in Supplementary Figure 8 (e.g., 1 is uncoloured metals, 2 is glazed ceramic, etc.). For each visual gloss feature (each panel) we show how the values for this feature were distributed for the original rendered stimuli in this material category (box plots, with ‘+’ symbols representing outliers), as well as the feature values for images obtained through our manipulation (colored lines). Here, the different colors represent the 4 shape-lightprobe combinations in our study, red: Dragon/Kitchen, magenta: Bunny/Kitchen, blue:

Dragon/Campus, green: Bunny/Campus). Three things are apparent from the plots: (1) visual features values obtained through image manipulation all pass through the box plots, suggesting that our manipulation indeed resulted in the appropriate feature values for each category; (2) overall, feature values are quite similar across shape-lightprobe combinations, yet some features vary slightly more across shape- lightprobe combinations, e.g. sharpness and coverage which are more affected by different lighting and geometry. We also applied the velvet filter in this condition (only for the velvet manipulation). Also see Supplementary Tables 2 and 3 for numerical values.'

Decision Letter, first revision:

Our ref: NATHUMBEHAV-210615734A

27th May 2022

Dear Dr. Schmid,

Thank you for submitting your revised manuscript "Material category of visual objects computed from specular image structure" (NATHUMBEHAV-210615734A). It has now been seen by the original referees and their comments are below. As you can see, the reviewers find that the paper has improved in revision (we also asked Reviewer #2 to comment on your responses to Reviewer #1's points and they were happy with them.) We will therefore be happy in principle to publish it in Nature Human Behaviour, pending minor revisions to satisfy the referees' final requests and to comply with our editorial and formatting guidelines.

We are now performing detailed checks on your paper and will send you a checklist detailing our editorial and formatting requirements within a week. Please do not upload the final materials and make any revisions until you receive this additional information from us.

Sincerely,

Jamie

Dr Jamie Horder
Senior Editor
Nature Human Behaviour

Reviewer #2 (Remarks to the Author):

Reviewer: Michael Landy

This is a pretty thorough revision and it certainly addresses all of my comments and desires for clarification. It's good to go!

Final Decision Letter:

Dear Dr Schmid,

We are pleased to inform you that your Article "Material category of visual objects computed from specular image structure", has now been accepted for publication in *Nature Human Behaviour*.

Please note that *Nature Human Behaviour* is a Transformative Journal (TJ). Authors whose manuscript was submitted on or after January 1st, 2021, may publish their research with us through the traditional subscription access route or make their paper immediately open access through payment of an article-processing charge (APC). Authors will not be required to make a final decision about access to their article until it has been accepted. IMPORTANT NOTE: Articles submitted before January 1st, 2021, are not eligible for Open Access publication. Find out more about Transformative Journals

With best regards,

Jamie

Dr Jamie Horder
Senior Editor
Nature Human Behaviour